# Structural and functional characterization of G protein–coupled receptors with deep mutational scanning

Eric M Jones[1†‡], Nathan B Lubock[1†‡], AJ Venkatakrishnan[2,3], Jeffrey Wang[1], Alex M Tseng[3], Joseph M Paggi[3], Naomi R Latorraca[3], Daniel Cancilla[1], Megan Satyadi[1], Jessica E Davis[1], M Madan Babu[2], Ron O Dror[3]*, Sriram Kosuri[1]*

[1]Department of Chemistry and Biochemistry, UCLA-DOE Institute for Genomics and Proteomics, Molecular Biology Institute, Quantitative and Computational Biology Institute, Eli and Edythe Broad Center of Regenerative Medicine and Stem Cell Research, and Jonsson Comprehensive Cancer Center, UCLA, Los Angeles, United States; [2]MRC Laboratory of Molecular Biology, Cambridge, United Kingdom; [3]Department of Computer Science, Stanford University, Department of Computer Science, Institute for Computational and Mathematical Engineering, Stanford University, Department of Computer Science, Department of Molecular and Cellular Physiology, Stanford University School of Medicine, Department of Computer Science, Department of Structural Biology, Stanford University School of Medicine, Stanford, United States

*For correspondence:
ron.dror@stanford.edu (ROD);
sri@ucla.edu (SK)

†These authors contributed equally to this work

Present address: ‡Octant, Inc, Emeryville, United States

**Abstract** The >800 human G protein–coupled receptors (GPCRs) are responsible for transducing diverse chemical stimuli to alter cell state- and are the largest class of drug targets. Their myriad structural conformations and various modes of signaling make it challenging to understand their structure and function. Here, we developed a platform to characterize large libraries of GPCR variants in human cell lines with a barcoded transcriptional reporter of G protein signal transduction. We tested 7800 of 7828 possible single amino acid substitutions to the beta-2 adrenergic receptor ($\beta_2$AR) at four concentrations of the agonist isoproterenol. We identified residues specifically important for $\beta_2$AR signaling, mutations in the human population that are potentially loss of function, and residues that modulate basal activity. Using unsupervised learning, we identify residues critical for signaling, including all major structural motifs and molecular interfaces. We also find a previously uncharacterized structural latch spanning the first two extracellular loops that is highly conserved across Class A GPCRs and is conformationally rigid in both the inactive and active states of the receptor. More broadly, by linking deep mutational scanning with engineered transcriptional reporters, we establish a generalizable method for exploring pharmacogenomics, structure and function across broad classes of drug receptors.

## Introduction

G-protein-coupled receptors (GPCRs) are central mediators of mammalian cells' ability to sense and respond to their environment. The >800 human GPCRs respond to a wide range of chemical stimuli such as hormones, odors, natural products, and drugs by modulating a small set of defined pathways that affect cellular physiology (*Isberg et al., 2016*; *Niimura et al., 2014*). Their central role in altering relevant cell states makes them ideal targets for therapeutic intervention, with ~34% of all U.S. Food and Drug Administration (FDA)-approved drugs targeting the GPCR superfamily (*Hauser et al., 2017*).

Understanding GPCR signal transduction is non-trivial for several reasons. First, GPCRs exist in a complex conformational landscape, making traditional biochemical and biophysical characterization difficult (*Deupi and Kobilka, 2010*; *Kobilka and Deupi, 2007*). Consequently, most experimentally determined GPCR structures are truncated, non-native, or artificially stabilized (*Isberg et al., 2016*). Even when structures exist, the majority are of inactive states - GPCR conformations that cannot couple with a G protein and cause it to stimulate intracellular signaling. Second, the function of a GPCR depends on its ability to change shape. Static structures from both X-ray crystallography and cryo electron microscopy do not directly probe structural dynamics (*Granier and Kobilka, 2012*). Tools such as double electron-electron resonance (DEER) spectroscopy, nuclear magnetic resonance (NMR) spectroscopy, and computational simulation have aided our understanding of GPCR dynamics, but interpreting how structural dynamics relate to function is still difficult (*Latorraca et al., 2017*; *Manglik and Kobilka, 2014*).

Structure- and dynamics-based analyses generate sets of candidate residues that are potentially critical for function and warrant further characterization. These approaches are complemented by methods that directly perturb protein function such as mutagenesis followed by functional screening. Several reporter gene and protein complementation assays measure GPCR signal transduction by activation of a transcriptional reporter, and are often used to identify and validate important structural residues (*Pei et al., 1994*; *Schönegge et al., 2017*; *Valentin-Hansen et al., 2012*). Such transcriptional reporter assays exist for most major drug receptor classes, including the major GPCR pathways: $G_{\alpha s}$, $G_{\alpha q}$, $G_{\alpha i/o}$, and arrestin signaling (*Azimzadeh et al., 2017*; *Cheng et al., 2010*; *Kroeze et al., 2015*).

Recent advances in DNA synthesis, genome editing, and next-generation sequencing have enabled deep mutational scanning (DMS) approaches that functionally assay all possible missense mutants of a given protein (*Fowler and Fields, 2014*; *Starita et al., 2017*). Several new methods allow for the generation and screening of DMS libraries in human cell lines and yeast (*Kotler et al., 2018*; *Lee et al., 2018*; *Majithia et al., 2016*; *Mavor et al., 2018*; *Starita et al., 2018*). Function is usually assessed by next-generation sequencing using screens that are bespoke to each gene's function, or by more general approaches that allow characterization of expression levels rather than function (*Matreyek et al., 2018*). For GPCRs, the DMS of the CXCR4, CCR5, and T1R2 GPCRs used binding to external epitopes to test expression and ligand binding (*Heredia et al., 2018*; *Park et al., 2019*). Unfortunately, such assays tell us little about the signaling capacity of these mutants, which is the primary function of GPCRs and many other drug receptors.

Here, we develop an experimental approach to simultaneously profile variant libraries with barcoded transcriptional reporters in human cell lines using RNA-seq. Methods to detect GPCR activation in multiplex have been previously described by us and others (*Botvinnik et al., 2010*; *Galinski et al., 2018*; *Jones et al., 2019*). Galinski et al.'s method reports on GPCR activity with a β-arrestin proximity sensor, requiring engineering of both arrestin and the GPCR, and enabling broad detection of GPCR activation across multiple signaling modalities. Our method is widely applicable to GPCRs and across the druggable genome where transcriptional reporters exist. As a proof-of-principle, we perform DMS on a prototypical GPCR, the β2-adrenergic receptor (β2AR) and measure the consequences of these mutations through the cyclic AMP (cAMP) dependent pathway, the primary signaling modality of Gs-coupled GPCRs.

## Results

### Multiplexed screening platform for G$_s$-coupled GPCR signaling

We developed a system to build, stably express, and assay individual variants of the β2AR in human cell lines. The β2AR primarily signals through the heterotrimeric G$_s$ protein, activating adenylyl cyclase upon agonist binding. In our platform, cAMP production stimulates transcription of a barcoded reporter gene, controlled by multimerized cAMP response elements (CRE, thus referred to as the CRE reporter for the rest of the manuscript), which can be quantified by RNA-seq (*Figure 1A*). Initially, we generated a HEK293T-derived cell line for stable integration of the GPCR-reporter construct (*Figure 1B*, *Figure 1—figure supplement 1A,B*). We also modified a previously developed Bxb1-landing pad system to allow for stable, once-only integration at the transcriptionally-silent H11 safe-harbor locus to avoid placing the CRE reporter within transcribed genes (*Cheung et al., 2019*;

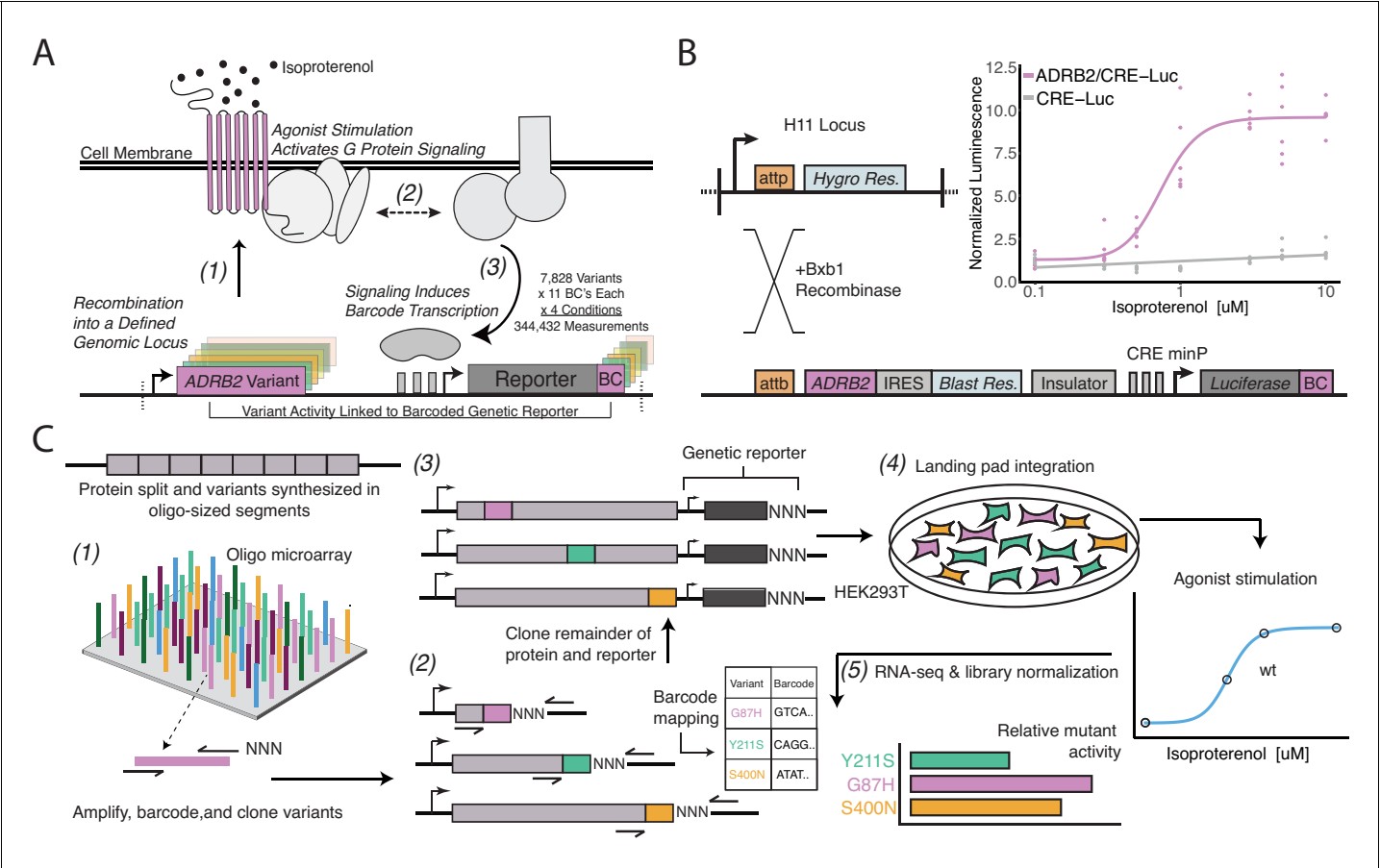

**Figure 1.** A platform for deep mutational scanning of GPCRs. (**A**) Overview of the multiplexed GPCR activity assay. Plasmids encoding *ADRB2* variants, a transcriptional CRE reporter of signaling activity, and 15 nucleotide barcode sequences that identify the variant are integrated into a defined genomic locus such that one variant is present per cell. Upon stimulation by isoproterenol, G-protein signaling induces transcription of the CRE genetic reporter and the barcode. Thus, the activity of a given variant is proportional to the amount of barcode mRNA which can be read out in multiplex by RNA-seq. (**B**) Schematic detailing the recombination of the reporter-receptor expression plasmid into the landing pad locus. Top right: activation of the CRE reporter integrated with (purple) or without (grey) exogenous *ADRB2* into the landing pad when stimulated with isoproterenol in Δ*ADRB2* cells via a luciferase CRE reporter gene assay. (**C**) Overview of library generation and functional assay. Missense variants are synthesized on an oligonucleotide microarray, the oligos are amplified with random DNA barcode sequences appended, and the variants are cloned into wild-type background vectors. Barcode-variant pairs are mapped with next-generation sequencing and the remaining wild-type receptor and CRE reporter sequences are cloned into the vector. Next, the variant library is integrated *en masse* into the serine recombinase (Bxb1) landing pad engineered at the H11 locus of Δ*ADRB2* HEK293T cells. This integration strategy ensures a single pair of receptor variant and barcoded CRE reporter is integrated per cell and avoids crosstalk. After selection, the library is stimulated with various concentrations of the β₂AR agonist, isoproterenol. Finally, mutant activity is determined by measuring the relative abundance of each variant's barcoded reporter transcript with RNA-seq.

The online version of this article includes the following figure supplement(s) for figure 1:

**Figure supplement 1.** Cellular engineering and reporter optimization for multiplexed assay.

*Duportet et al., 2014*; *Matreyek et al., 2017*). To prevent endogenous signaling, we knocked out the gene encoding for β₂AR, *ADRB2,* and verified loss of CRE reporter gene activity in response to the β₂AR agonist, isoproterenol (*Figure 1—figure supplement 1C*). Our donor vector configuration ensures the receptor and resistance marker are only activated upon successful integration into the landing pad (*Figure 1B*). Lastly, we included several sequence elements in the donor vector to improve signal-to-noise of the assay: an insulator upstream of the CRE reporter and an N-terminal affinity tag to the receptor (*Figure 1—figure supplement 1D,E*). As a result, upon integration of a donor vector expressing wild-type (WT) β₂AR, isoproterenol induces CRE reporter gene expression in a dose-dependent manner (*Figure 1B*).

We designed and synthesized the receptor's 7828 possible missense variants in eight segments on oligonucleotide microarrays (*Figure 1C*). We amplified the mutant oligos, attaching a random 15 nucleotide barcode sequence, and cloned them into one of eight background vectors encoding the upstream, wild-type portion of the gene. In this configuration, we mapped barcode-variant pairs with next-generation sequencing and subsequently utilized Type IIS restriction enzymes to insert the remaining sequence elements between the receptor and barcode. In the resulting mature donor vector, the barcode is located in the 3' untranslated region (UTR) of the CRE reporter gene. We integrated the library into our engineered cell line, and developed protocols to ensure proper quantification of library members, most notably vastly increasing the numbers of cells we assayed and RNA processed for the RNA-seq (*Figure 1—figure supplement 1F*, *Figure 2—figure supplement 1A*).

## Measurement of mutant activities and comparison to evolutionary metrics

We screened the mutant library at four concentrations of the $\beta_2$AR full-agonist isoproterenol: vehicle control, an empirically determined half-maximal activity ($EC_{50}$, 150 nM), full activity ($EC_{100}$, 625 nM), and beyond saturation of the WT receptor ($E_{max}$, 5 µM). We obtained reliable measurements (coefficient of variation <1) for 95–99% (7,461–7,749/7,828 depending on the agonist concentration) of possible missense variants (412 residues * 19 amino acids = 7828 possible missense variants) with two biological replicates at each condition (*Figure 1C*). We normalized these measurements against forskolin treatment, which induces cAMP signaling independent of the $\beta_2$AR (*Insel and Ostrom, 2003*). Forskolin treatment maximally induces the CRE reporter gene, therefore the relative barcode expression is proportional to the physical composition of the library. Each cell contains a single copy of the same CRE reporter sequence, therefore any differences in maximum transcriptional output between barcodes will be due to differences in the frequency of each barcode within the cell library. Finally, we define activity as the ratio of this value to the mean frameshift (Materials and methods). Each variant was represented by 10 barcodes (median), with biological replicates displaying Pearson's correlations of 0.87 to 0.90 at the barcode level and 0.66 to 0.75 when summarized by individual variants (*Figure 1—figure supplement 1G,H*, *Figure 2—figure supplement 1A*). Of note, we aimed for 10 barcodes per variant in order to account for any effects individual barcodes will have on CRE reporter transcription and serve as statistical replicates for each variant.

The heatmap representation of the variant-activity landscape reveals global and regional trends in response to specific subtypes of mutations (*Figure 2A*). For example, the transmembrane domain and intracellular helix eight are more sensitive to substitution than the termini or loops, and this effect becomes more pronounced at higher agonist concentrations (*Figure 2A*; all p<0.001; Mann-Whitney U). The transmembrane domain and intracellular helix eight are also sensitive to helix-disrupting proline substitutions (*Figure 2B*, *Figure 2—figure supplement 1B*; all p<<0.001 except TM vs Helix-8; Mann-Whitney U). Microarray-derived DNA often contains single-base deletions (47% of oligos in our library) that will introduce frameshift mutations into our library (*LeProust et al., 2010*). As expected, frameshifts consistently display lower activity than missense mutations regardless of agonist concentration (*Figure 2C*; p<<0.001; Mann-Whitney U). Furthermore, the effect of frameshifts are markedly decreased in the C-terminus of the protein (*Figure 2D*; p<<0.001; Mann-Whitney U). We also built and integrated previously characterized mutants (*Elling et al., 1999*; *Sato et al., 1999*; *Shenoy et al., 2006*) into our system individually and measured activity with a luciferase CRE reporter gene at the same induction conditions (*Figure 2E* and *Figure 2—figure supplement 1C*). As expected, known null mutations (D113A and I135W) have significantly diminished activity relative to WT in both systems, even at $E_{max}$ (all p<<0.001; Wald Test). Known hypomorphic mutations (S203A and S204A) also have a significant decrease in activity relative to WT at $EC_{100}$ (all p<<0.001; Wald Test), but are not significantly different than WT at $E_{max}$ as expected (all p>0.01; Wald Test).

Metrics for sequence conservation and covariation are often used to predict the effects a mutation will have on protein function (*Adzhubei et al., 2013*; *Capra and Singh, 2007*; *Hopf et al., 2017*). Mutational tolerance, the mean activity of all amino acid substitutions per residue at each agonist concentration, is highly correlated to conservation, both across species for the $\beta_2$AR (*Figure 3—figure supplement 1A*; Spearman's $\rho = -0.74$; 55 orthologs, predominantly mammals but including a few other vertebrates as well as a small number of invertebrate beta-like sequences, identified from the OMA Database, *Supplementary file 1*), and across all Class A GPCRs

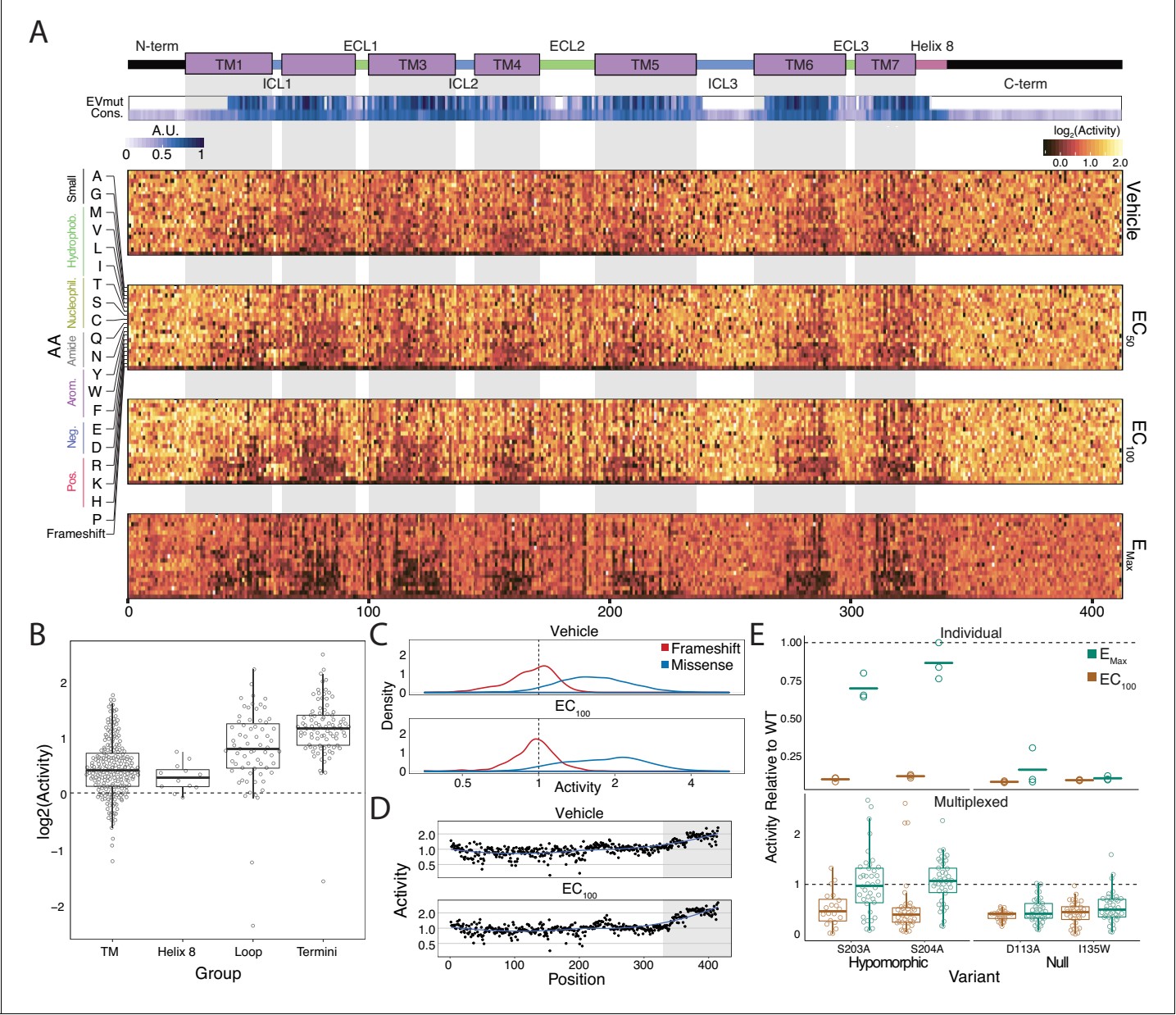

**Figure 2.** Variant-activity landscape for 7800 missense variants of the β2AR and multiplexed assay validation. (**A**) Top: Secondary structure diagram of the β$_2$AR: the N and C termini are black, the transmembrane helices are purple blocks, and the intra- and extracellular domains are colored blue and green, respectively. The EVmutation track (EVmut.) displays the mean effect of mutation at each position as predicted from sequence covariation (*Hopf et al., 2017*). Conservation track (Cons.) displays the sequence conservation of each residue across 55 β$_2$AR orthologs from the OMA database (*Capra and Singh, 2007*; *Altenhoff et al., 2018*). A.U. stands for arbitrary units and the scale for the EVmutation and sequence conservation tracks are individually 0–1 normalized. The shaded guides represent positions in the transmembrane domain. Bottom: The heatmap representation of mutant activity at each agonist condition. Variants are colored by their activity score. relative to the mean frameshift mutation. Activity is the measurement of signaling for each variant relative to the mean frameshift (see methods). (**B**) The distribution of mutant activity for proline substitutions is significantly different for amino acids that reside in the transmembrane domain/helix eight to those in the flexible loops and termini at EC$_{100}$ (all p<<0.001 except TM vs Helix 8; Mann-Whitney U). (**C**) The distribution of frameshift mutant activity (red) is significantly different than the distribution of designed missense mutations (blue) across increasing isoproterenol concentrations (both p<<0.001; Mann-Whitney U). Mean frameshift activity marked with a dashed line. (**D**) Relative effect of the mean frameshift mutant activity per position is markedly decreased in the unstructured C-terminus of the protein (shaded region) and is consistent across agonist concentration (both p<<0.001; Mann-Whitney U). Blue line represents the LOESS fit. (**E**) Mutant activity measured individually with a luciferase CRE reporter gene compared to the multiplexed assay at EC$_{100}$ and E$_{Max}$ isoproterenol induction. Known null mutations (D113A, I135W) have no dose response between EC$_{100}$ and E$_{max}$ and are significantly different than synonymous mutants at both concentrations in both systems (all p<<0.001; Wald test). Alternatively, known hypomorphic mutations (S203A, S204A) are significantly different than

*Figure 2 continued on next page*

*Figure 2 continued*

synonymous mutations at $EC_{100}$ (all p<<0.001; Wald test), but are not significantly different at $E_{max}$ (all p>0.01; Wald test). Bars represent mean value in the luciferase data. In the Individual facet, each dot represents a replicate measurement and in the multiplexed facet, each dot represents a different barcode.

The online version of this article includes the following figure supplement(s) for figure 2:

**Figure supplement 1.** Global metrics of the multiplexed screen.

(Spearman's ρ = −0.68; *Figure 3A* and *Figure 3—figure supplement 1B*; *Altenhoff et al., 2018*; *Capra and Singh, 2007*; *Hopf et al., 2017*) at $EC_{100}$. From this point on, any use of the words tolerance or intolerance in this manuscript refer to mutational tolerance. Correlation between our data and both predictors increases with agonist concentration up to $EC_{100}$ (*Figure 3—figure supplement 1A,B*). We found a subset of residues in extracellular loop 2 (ECL2), including C184 and C190 that form an intraloop disulfide bridge, that were more intolerant to mutation than expected given their conservation across Class A GPCRs. This suggests a fairly specific functional role for this motif in the β₂AR (*Figure 3A*). On an individual variant level, mutational responses correlate (Spearman's ρ = 0.520) with EVmutation, a predictor of mutational effects from sequence covariation (*Figure 3B* and *Figure 3—figure supplement 1C*; *Altenhoff et al., 2018*; *Capra and Singh, 2007*; *Hopf et al., 2017*).

## Population genetics and structural analysis of individual variants

In addition to evolutionary metrics, understanding the functional distribution of *ADRB2* variants found within the human population is important given the extensive variation found among GPCR drug targets (*Hauser et al., 2018*). The Genome Aggregation Database (gnomAD) reports variants found across 141,456 individuals (*Karczewski et al., 2019*), and many of the 180 *ADRB2* missense variants are of unknown significance. We classified 11 of these variants as potentially loss of function, by comparing their activity to the distribution of frameshift mutations found in our assay (*Figure 3C*; see Materials and methods). Given that measurements of individual mutations are noisy (average coefficient of variation = 0.55), this analysis is best suited as a funnel to guide further characterization (see Discussion).

However, our analysis is more robust when we aggregate the signal of multiple mutations at a given position. Therefore, we compiled a list of the 100 most activating mutations at vehicle control and the 100 least active mutations at $EC_{100}$ and mapped their location on the β₂AR structure. As expected, the least active mutations tended to reside within the core of the transmembrane domain (*Figure 3—figure supplement 1D,E*). Alternatively, the most activating mutations mapped to TM1, TM5, TM6, and residues that typically face away from the internal core of the receptor (*Figure 3D, E*). Of note, a group of these mutations in TM5 face TM6, which undergoes a large structural rearrangement during receptor activation (*Weis and Kobilka, 2018*). Activating mutants are also enriched in the termini, ICL3, and Helix 8. Concentration at the termini is unsurprising, as these regions have known involvement in surface expression and our current assay does not discriminate between increased signaling potency and expression (see discussion; *Dong et al., 2007*). However, there are cases of constitutively active mutations in the N terminus that increase signaling potency without affecting surface expression, such as T11S of the melanocortin 4 (MC4R) (*Lotta et al., 2019*). Similarly, the enrichment of activating mutants in ICL3 appears to reflect its role in G-protein binding (*Ozcan et al., 2013*; *Ozgur et al., 2016*; *West et al., 2011*). Lastly, we observe a number of activating mutations in the terminal residue, L413. A recent study of genetic variation in human *MC4R* also found a gain-of-function mutation at the terminal residue of the receptor, suggesting a possible conserved role for this position in regulating basal activity of GPCRs (*Lotta et al., 2019*).

## Unsupervised learning reveals functionally relevant groupings of residues

Given that our data spans thousands of mutations across several treatment conditions, we used unsupervised learning methods to reveal hidden regularities within groups of residues' response to

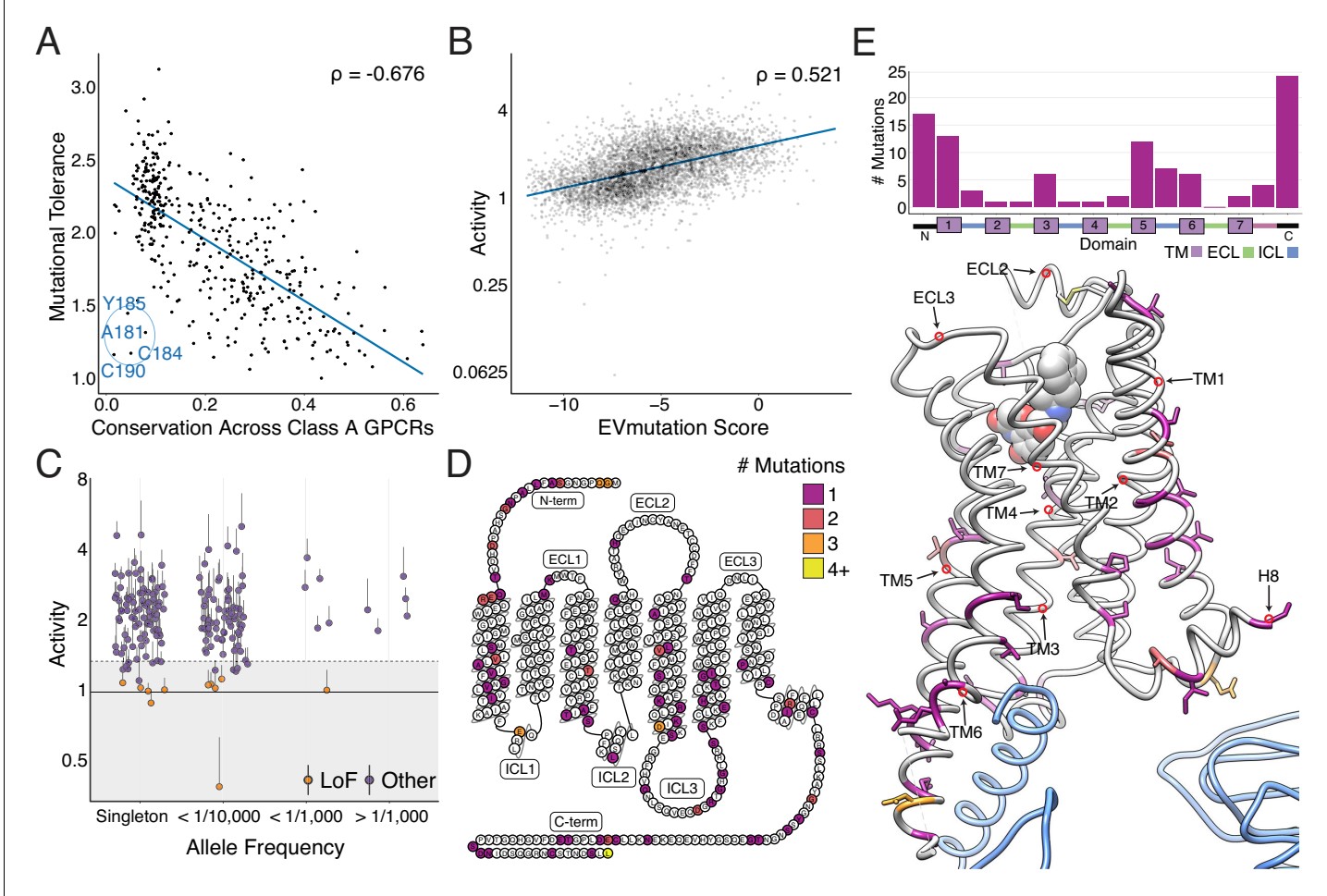

**Figure 3.** Individual mutations and residues reveal evolutionary and structural insights into β2AR function. (A) Positional conservation across Class A GPCRs correlates with mutational tolerance (Spearman's $\rho = -0.676$, Pearson's $r = -0.681$), the mean activity of all amino acid substitutions per residue at each agonist concentration, at $EC_{100}$. However, four of the least conserved positions (C190, C184, A181, Y185) are highly sensitive to mutation and are located in ECL2, suggesting this region is uniquely important to the β2AR. The blue line is a simple linear regression. (B) Individual mutant activity correlates with EVmutation (Spearman's $\rho = 0.521$, Pearson's $r = 0.480$) at $EC_{100}$. The blue line is a simple linear regression. (C) Activity of individual mutants present in the human population from the gnomAD database stratified by allele frequency. Mutations are classified as potential loss of function (LoF) mutations (orange) are classified as such (shaded region) if the mean activity at $EC_{100}$ plus the standard error of the mean (upper error bar) is more than two standard deviations below mean frameshift mutant activity (dashed line). (D) The distribution of the 100 most basally activating mutations across the β2AR snake plot reveals a clustering of mutants in the termini, TM1, TM5, and TM6. (E) Top: Distribution of the 100 most basally activating mutations stratified by domain. Bottom: The distribution of the 100 most basally activating mutations across the β2AR 3D structure (PDB: 3SN6). These positions (colored as in D) are concentrated on the surface of the β2AR ($G_{\alpha s}$ shown in blue).

The online version of this article includes the following figure supplement(s) for figure 3:

**Figure supplement 1.** Correlation with sequence conservation and covariation and analysis of individual mutations.

mutation. In particular, we applied Uniform Manifold Approximation and Projection (UMAP) (*McInnes and Healy, 2018*) to learn multiple different lower dimensional representations of our data and clustered the output with density-based hierarchical clustering (HDBSCAN; *Figure 4— figure supplement 1*; *Campello et al., 2013*). We found residues consistently separated into six clusters that exhibit distinct responses to mutation (*Figure 4A,B*). Clusters 1 and 2 are globally intolerant to all substitutions, whereas Cluster 3 is vulnerable to proline and charged substitutions. Cluster 4 is particularly inhibited by negatively charged substitutions and Cluster five by proline substitutions, while Cluster 6 is unaffected by any mutation. Mapping these clusters onto a 2D snake plot representation shows Clusters 1–5 primarily comprise the transmembrane domain, while Cluster

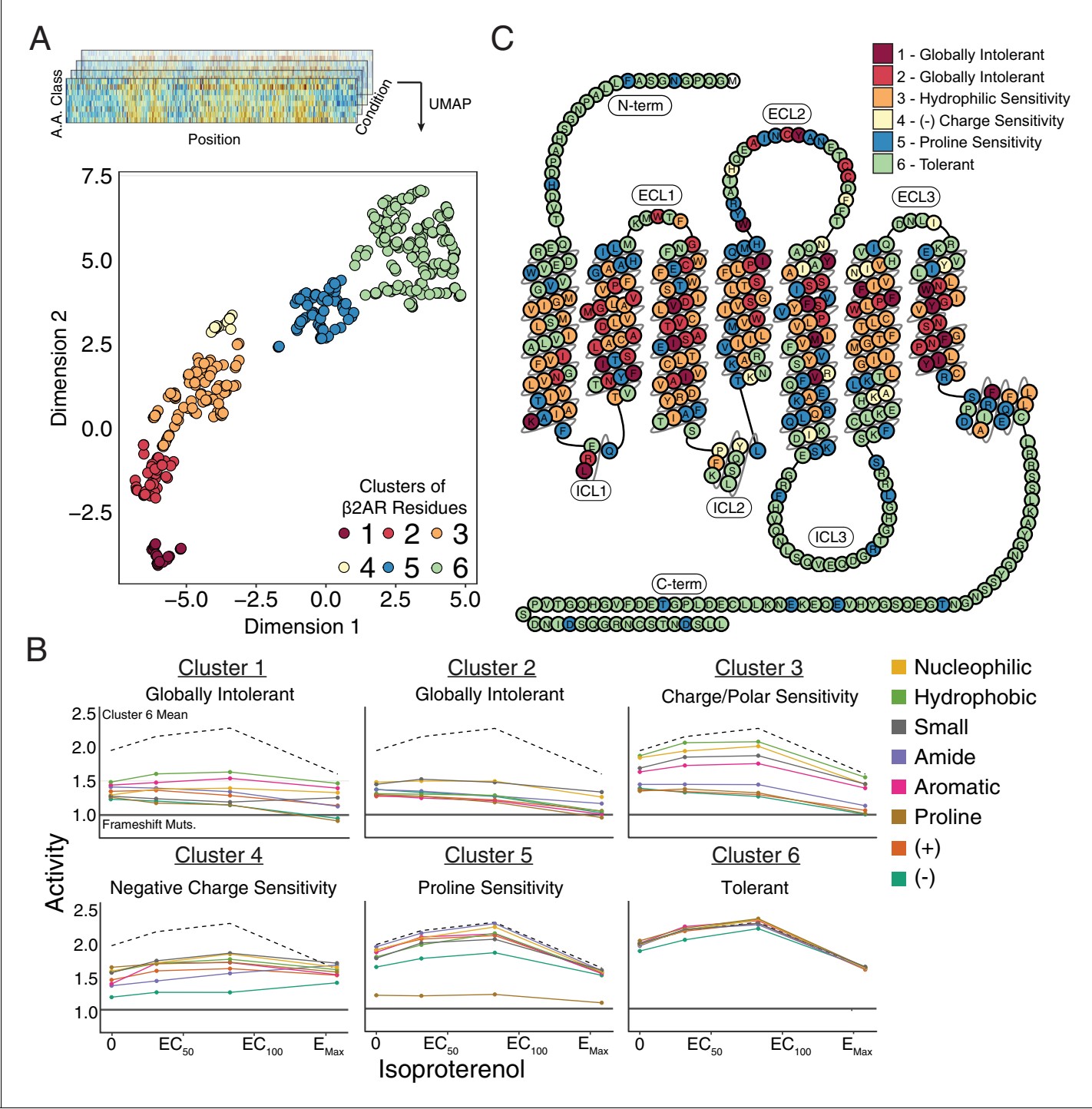

**Figure 4.** Unsupervised learning segregates residues into clusters with distinct responses to mutation. (A) Amino acids were segregated into classes based on their physicochemical properties and mean activity scores were reported by class for each residue. With Uniform Manifold Approximation and Projection (UMAP) a 2D representation of every residue's response to each mutation class across agonist conditions was learned. Each residue is assigned into one of six clusters using HDBSCAN (see *Figure 4—figure supplement 1*). (B) Class averages for each of these cluster reveal distinct responses to mutation. The upper dashed line represents the mean activity of Cluster 6 and the lower solid line represents the mean activity of frameshift mutations. (C) A 2D snake plot representation of β2AR secondary structure with each residue colored by cluster identity.

The online version of this article includes the following figure supplement(s) for figure 4:

**Figure supplement 1.** Cluster assignment is robust across different UMAP embeddings.

6 resides in the loops and termini (*Figure 4C*). These flexible regions are often truncated before crystal structure determination to minimize conformational variability (*Rosenbaum et al., 2007*). Surprisingly, a number of residues from Cluster five also map there, suggesting potential structured regions. However, Cluster 5 assignment is largely based on the response of a single proline mutation, and thus is more susceptible to noise than the other clusters (see Discussion).

Next, we projected the clusters onto the hydroxybenzyl isoproterenol-bound structure (*Figure 5— figure supplement 1A*; PDB: 4LDL). The globally intolerant Clusters 1 and 2 segregate to the core of the protein, while the charge-sensitive Cluster 3 is enriched in the lipid-facing portion (*Figure 5— figure supplement 1B*). This suggests that differential patterns of response to hydrophobic and charged substitutions could correlate with side chain orientation within the transmembrane domain. Indeed, residues that are uniquely charge sensitive are significantly more lipid-facing than those that are sensitive to both hydrophobic and charged mutations at $EC_{100}$ (*Figure 5A*, *Figure 5—figure supplement 1C–D*, p=0.000036; Mann-Whitney U) (*Mitternacht, 2016*).

## Mutational tolerance stratifies the functional relevance of structural features

Decades of research have revealed how ligand binding is coupled to G-protein activation through a series of conserved motifs (*Weis and Kobilka, 2018*). This comprehensive, unbiased screen enables us to systematically evaluate and rank the functional importance of every implicated residue. The globally intolerant UMAP clusters (1 and 2) highlight many residues from these motifs and suggest novel residues for investigation (*Figure 5B*). We can further resolve the significance of individual residues within these motifs by ranking the mutational tolerance of positions in these clusters at $EC_{100}$ (*Figure 5C*). In fact, 11 of the 15 most mutationally intolerant positions belong to the PIF, CWxP, and NPxxY motif, orthosteric site, a water-mediated bond network, an extracellular disulfide bond, and a cholesterol-binding site. Interestingly, the second most intolerant residue is the uncharacterized G315[7x41] (GPCRdb numbering in superscript *Isberg et al., 2016*). In the active state, G315's alpha carbon points directly at W286[6x48] of the CWxP motif, the fourth most intolerant residue, and any substitution at G315[7.x41] will likely clash with W286[6x48] (*Figure 5D*). We confirmed G315's intolerance with a luciferase CRE reporter gene assay, where mutants G315T and G315L resulted in complete loss of function (*Figure 5—figure supplement 2A*).

Recent simulations suggest water-mediated hydrogen bond networks play a critical role in GPCR function (*Venkatakrishnan et al., 2018*; *Venkatakrishnan et al., 2019*). The third most intolerant residue in our assay, Y326[7x53] of the NPxxY motif, is especially important as it switches between two of these networks during receptor activation. In the inactive state, Y326[7x53] contacts N51[1x50] and D79[2x50], two of the top 15 most intolerant positions (*Figure 5E*). N51L and N51Y also result in complete loss of function when assayed individually (*Figure 5—figure supplement 2A*). The movement of Y326[7x53] is also part of a broader rearrangement of residue contacts that are conserved across Class A GPCRs, with the majority of these residues being intolerant to mutation (*Figure 5—figure supplement 2B*; *Venkatakrishnan et al., 2016*). Aside from G315[7x41], the other uncharacterized residues in the top 15 include W99[23x50], S319[7x46], and G83[2x54]. Given the correlation between mutational tolerance and functional relevance, further investigation of these residues will likely reveal insights into GPCR biology.

Next, we hypothesized residues in the orthosteric site that directly contact isoproterenol would respond uniquely to mutation; however, no crystal structure of $\beta_2AR$ bound to isoproterenol exists. Using the crystal structure of the $\beta_2AR$ bound to the analog, hydroxybenzyl isoproterenol (PDB: 4LDL), we find that residues responsible for binding the derivatized hydroxybenzyl tail have significantly higher mutational tolerance than residues that contact the catecholamine head common to both isoproterenol and hydroxybenzyl isoproterenol at $EC_{100}$ (p=0.0162; *Figure 5F*, *Figure 5—figure supplement 2C*). Given this discrimination, we believe DMS can be a powerful tool for mapping functional ligand-receptor contacts in GPCRs.

GPCR signaling is dependent on a series of intermolecular interactions, and the numerous $\beta_2AR$ crystal structures enable us to comprehensively evaluate residues mediating such interactions. For example, cholesterol is an important modulator of GPCR function (*Thal et al., 2018*), and the timolol-bound inactive-state $\beta_2AR$ structure elucidated the location of a cholesterol-binding site (PDB: 3D4S) (*Hanson et al., 2008*). Of residues in this pocket, W158[4x50] is predicted to be most important for cholesterol binding, and in agreement, W158[4x50] is the most mutationally intolerant (*Figure 5—*

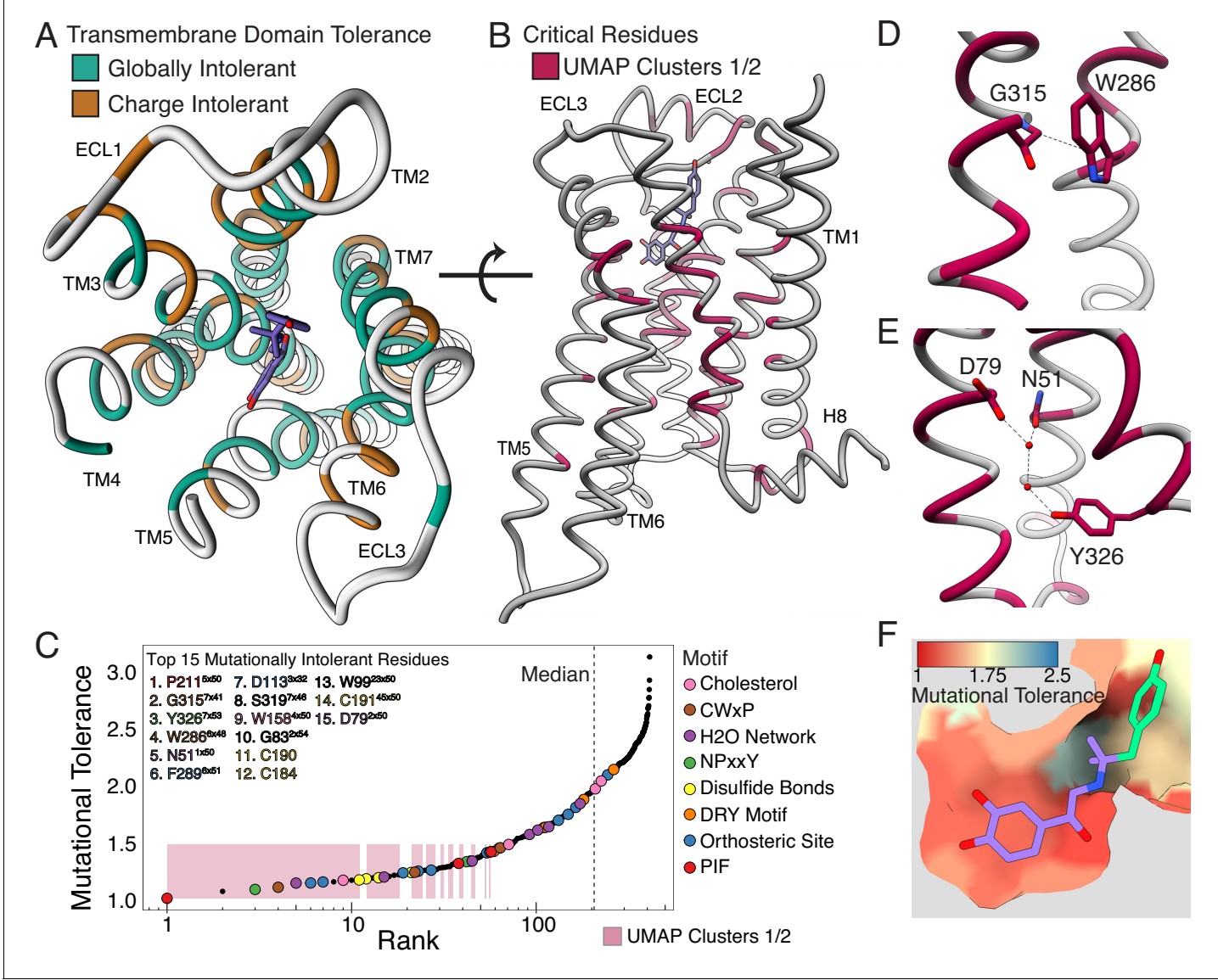

**Figure 5.** Mutational tolerance elucidates broad structural features and critical residues of the $\beta_2AR$. (A) Residues within the transmembrane domain colored by their tolerance to particular classes of amino acid substitution. Teal residues are intolerant to both hydrophobic and charged amino acids (globally intolerant), and brown residues are tolerant to hydrophobic amino acids but intolerant to charged amino acids (charge intolerant). The charge-sensitive positions' side chains are enriched pointing into the membrane, while the globally intolerant positions' side chains face into the core of the protein (see *Figure 5—figure supplement 1*). (B) The crystal structure of the hydroxybenzyl isoproterenol-activated state of the $\beta_2AR$ (PDB: 4LDL) with residues from the mutationally intolerant Clusters 1 and 2 highlighted in maroon. (C) 412 $\beta_2AR$ residues rank ordered by mutational tolerance at the $EC_{100}$ isoproterenol condition. Residues in known structural motifs (colored points) are significantly more sensitive to mutation than other positions on the protein ($p \ll 0.001$). Dashed line demarcates the median of the ranking. The top 15 mutationally intolerant residues are listed and colored by motif association. (D-F) Selected vignettes of residues from the mutationally intolerant UMAP clusters and ranking. (D) W286$^{6x48}$ of the CWxP motif and the neighboring G315$^{7x41}$ are positioned in close proximity. Substitutions at G315$^{7x41}$ are likely to cause a steric clash with W286$^{6x48}$ (PDB: 4LDL). (E) An inactive-state water-mediated hydrogen bond network (red) associates N51$^{1x50}$ and Y326$^{7x53}$ (PDB: 2RH1). Disruption of this network may destabilize the receptor. (F) The ligand-bound orthosteric site surface colored by mutational tolerance. Receptor-ligand contacts with the catecholamine head (present in agonist used in assay) are more intolerant to mutation than those in the hydroxybenzyl tail (not present in agonist used in assay) of the isoproterenol analog depicted in this crystal structure (PDB: 4LDL).

The online version of this article includes the following figure supplement(s) for figure 5:

**Figure supplement 1.** Mutational profile suggests side chain orientation and environment.

**Figure supplement 2.** Mutational intolerance of functionally related residues.

*figure supplement 2D*). Similarly, a number of studies have mutagenized residues at the $G_{\alpha s}$-$\beta_2$AR interface (*Jensen et al., 2001*; *Moro et al., 1993*; *O'Dowd et al., 1988*; *Rasmussen et al., 2011*; *Sheikh et al., 1999*; *Swaminath et al., 2003*; *Valentin-Hansen et al., 2012*; *Valiquette et al., 1995*), but a complete understanding of the relative contribution of each residue to maintaining the interface is unknown. Most residues are more mutationally tolerant than residues in the intolerant Clusters 1 and 2, but the four most intolerant positions are I135$^{3x54}$, V222$^{5x61}$, A271$^{6x33}$, and Q229$^{5x68}$ respectively (*Figure 5—figure supplement 2E*). Q229$^{5x68}$ appears to coordinate polar interactions between D381 and R385 of the $\alpha5$ helix of $G_{\alpha s}$, whereas V222$^{5x61}$ and I135$^{3x54}$ form a hydrophobic pocket on the receptor surface (*Figure 5—figure supplement 2F*).

## A structural latch is conserved across Class A GPCRs

Analysis of the mutational tolerance data has highlighted the functional importance of previously uncharacterized residues. In particular, W99$^{23x50}$ of extracellular loop 1 (ECL1) is the 13[th] most intolerant residue, which is unusual as mutationally intolerant residues are rare in the flexible loops. Furthermore, W99$^{23x50}$ is proximal to the disulfide bond C106$^{3x25}$-C191$^{45x50}$, an important motif for stabilization of the receptor's active state (*Noda et al., 1994*; *Dohlman et al., 1990*; *Hulme, 2013*; *Dohlman et al., 1990*; *Noda et al., 1994*). While aromatic residues are known to facilitate disulfide bond formation, only tryptophan is tolerated at this position (*Bhattacharyya et al., 2004*). We hypothesize W99's indole group hydrogen bonds with the backbone carbonyl of the neighboring uncharacterized and mutationally intolerant G102$^{3x21}$, positioning W99$^{23x50}$ toward the disulfide bond. Other aromatic residues are unable to form this hydrogen bond and are less likely to be positioned properly. G102$^{3x21}$ also hydrogen bonds with the backbone amide of C106$^{3x25}$, further stabilizing this region. To verify this claim, we individually confirmed the mutational intolerance of both W99$^{23x50}$ and G102$^{3x21}$ (*Figure 6—figure supplement 1A*). Additionally, we evaluated surface expression for a subset of W99$^{23\times50}$ and G102$^{3\times21}$ mutants (*Figure 6—figure supplement 1B*). Relative to three previously characterized mutants with severely impaired surface expression (*Parmar et al., 2017*) and wild-type β2AR, the mutants exhibited mildly impaired to normal surface expression—supporting a role in signaling for these residues.

Interestingly, W99$^{23x50}$, G102$^{3x21}$, and C106$^{3x25}$ are almost universally conserved across Class A GPCRs (*Vass et al., 2018*; *Figure 6A*, *Figure 6—figure supplement 1C*). Comparison of over 25 high-resolution structures of class A GPCRs from five functionally different sub-families and six different species revealed that these residues consistently contact each other (*Figure 6B,C*). Based on the evolutionary and structural conservation across Class A GPCRs, we find W99$^{23x50}$, G102$^{3x21}$, and the C106$^{3x25}$-C191$^{45x50}$ disulfide bond represent a conserved WxxGxxxC motif, forming an extracellular 'structural latch' that is maintained consistently throughout GPCRs spanning diverse molecular functions and phylogenetic origins. While a minority of Class A GPCRs lack the Trp/Gly combination of residues in the ECL1 region, these receptors have varying structures in ECL1: an alpha helix (sphingosine S1P receptor), beta strand (adenosine receptor), or even intrinsically disordered (viral chemokine receptor US28) (*Figure 6—figure supplement 1D*).

To better understand the dynamics of the structural latch, we compared the active and inactive state crystal structures of four representative GPCRs. While the overall RMSD between the inactive and active states for the β2AR, M2 muscarinic receptor, and μ opioid receptorreceptor are 1 Å,1.5 Å, and 1.7 Årespectively, the conformation of the latch in the active and inactive states is nearly identical in each receptor (*Figure 6D*). This suggests that the extracellular structural latch is part of a larger rigid plug present at the interface of the transmembrane and extracellular regions, which could be important for the structural integrity of the receptor and possibly guide ligand entry.

In Class A receptors lacking components of the WxxGxxxC motif, introducing the Trp-Gly interaction could increase the stability of the receptor for structural studies. In fact, in the BLT1 receptor structure, a Gly mutation at 3×21 was found to be thermostabilizing (*Hori et al., 2018*). Other candidate receptors lacking a Gly at 3×21 include the alpha2B receptor and the neuropeptide FF2 receptor, where the R81G and D112G mutations have potential to increase receptor stability, respectively. More broadly, these ECL1/TM3 positions conserved across Class A GPCRs could serve as candidate sites for introducing thermostabilizing mutations.

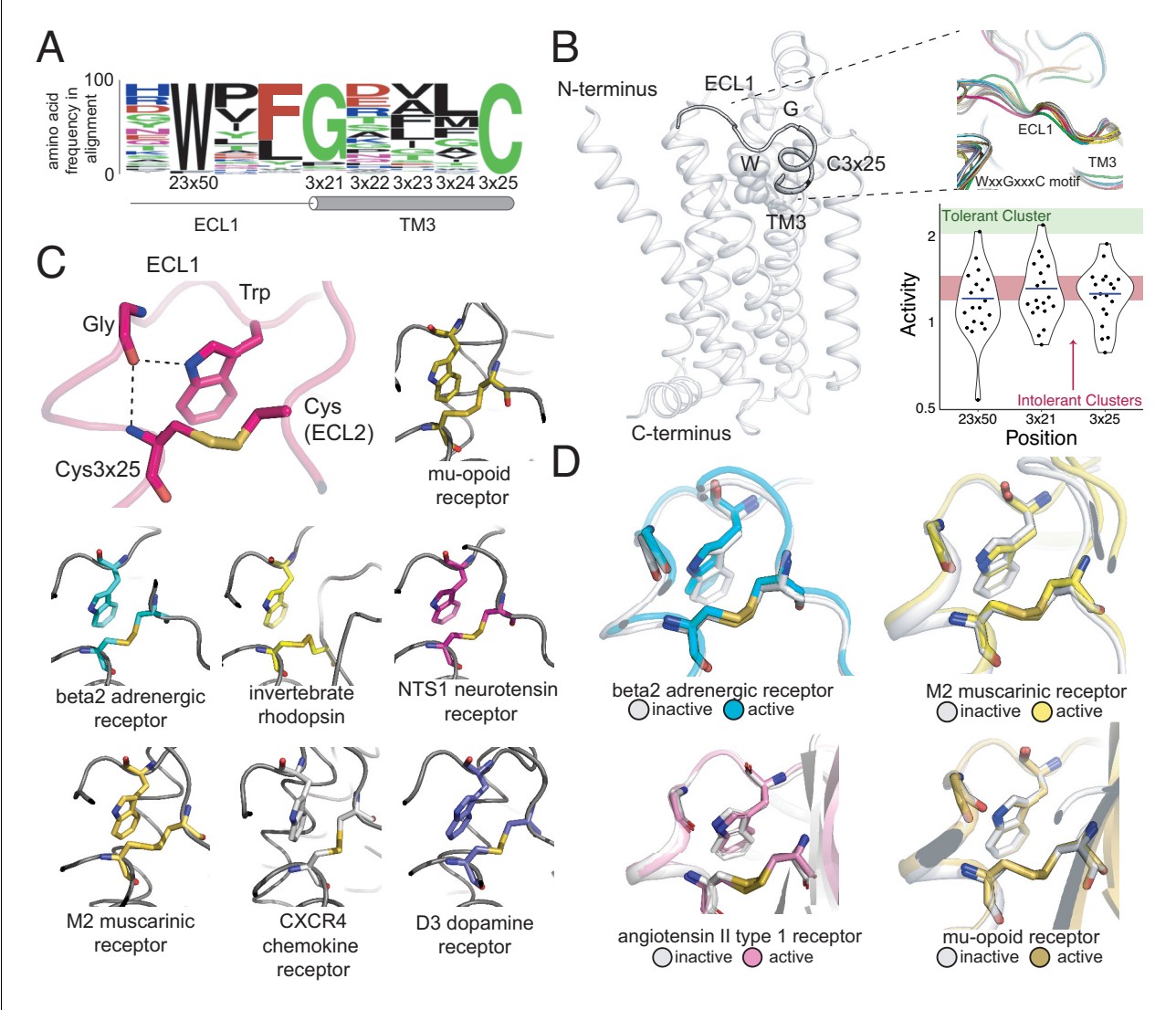

**Figure 6.** A conserved extracellular tryptophan-disulfide 'structural latch' in class A GPCRs is mutationally intolerant and conformation-independent. (A) Sequence conservation of extracellular loop 1 (ECL1) and the extracellular interface of TM3 (202 Class A GPCRs with a disulfide bridge between TM3 and ECL1). (B) Left: Depiction of the interaction of W99$^{23x50}$, G102$^{3x21}$, and C106$^{3x25}$ in ECL1 of the β2AR. Top Right: Conservation of the structure of the ECL1 region across functionally different class A GPCRs. Bottom Right: Activity of all 19 missense variants assayed for each of the three conserved residues, with the mean activity (mutational tolerance) shown as a blue bar. The shaded bars represent the mean mutational tolerance ± 1 SD of residues in the tolerant Cluster 6 (green) and the intolerant Clusters 1 and 2 (red). (C) A hydrogen bond network between mutationally intolerant positions W99$^{23x50}$, G102$^{3x21}$, and C106$^{3x25}$. Representative examples of the structural latch are shown. (D) This structural latch is maintained in both the inactive and active state structures for the β2AR (inactive: 2RH1, active: 3P0G), the M2 muscarinic receptor (inactive: 3UON, active: 4MQS), the angiotensin II type one receptor (inactive: 4ZUD, active: 6OS1), and the mu-opioid receptor (inactive: 4DKL, active: 5C1M).

The online version of this article includes the following figure supplement(s) for figure 6:

**Figure supplement 1.** The WxxGxxxC motif is highly conserved across Class A GPCRs.

## Discussion

Our findings showcase a new, generalizable approach for DMS of human protein targets with transcriptional reporters. Such reporters enable precise measurements of gene-specific function that can be widely applied across the druggable genome. We show comprehensive mutagenesis can illuminate the structural organization of the protein and the local environment of individual residues. These results suggest DMS can work in concert with other techniques (e.g. X-ray crystallography, Cryo-EM, and molecular dynamics) to augment our understanding of GPCR structure-function

relationships. Moreover, we identify key residues for β2AR function including uncharacterized positions that inform about receptor stability and activation. Importantly, these approaches can be undertaken when direct structural information is unavailable but reporters exist, which is true for most GPCRs.

There are still a number of limitations to our current approach that we expect will improve as we develop the method. Importantly, we did not quantify cell-surface expression directly in our high-throughput functional assay, and thus we cannot distinguish between mutations that substantially affect G-protein signaling and those that affect cell-surface expression. In particular, mutations that lead to increased signal in our assays could in fact work by reducing GPCR internalization and not by increasing the intrinsic activity of the receptor. However, we express our variant library in a genomic context at a controlled copy number, dampening the effects of expression-related artifacts typically associated with assays that involve transiently transfected receptor. In addition, expression level alterations can affect the dynamics of signaling and thus may be physiologically relevant. For example, the GPCR *MC4R* is haploinsufficient, and rare heterozygous mutations that eliminate or reduce receptor expression are associated with obesity (*Farooqi et al., 2003*; *Khera et al., 2019*; *Lotta et al., 2019*). Combining our assay with new generalized, multiplexed assays of protein expression levels in human cells can help tease apart mechanistic reasons for differences in signaling (*Matreyek et al., 2018*). Secondly, the current signal-to-noise ratio of this approach at single-variant resolution restricted our analyses to mutations with extreme effects on receptor function. This made interpreting single mutations challenging. For example, several mutations within the C terminus exhibited a sensitivity to proline substitution. This was surprising because the C terminus is thought to be a flexible, disordered region (*Cherezov et al., 2007*; *Rasmussen et al., 2007*). We individually synthesized and tested three of these mutations (E369P, R253P, and T360P) and found that they did not disrupt function (*Figure 5—figure supplement 2A*). Thus, individual variant data should be confirmed by more traditional assays until the signal-to-noise ratio is improved. However, our measurements are robust in aggregate, and pointed to new receptor biology, providing structural and functional insights. Further improvements to the signal-to-noise will facilitate the exploration of more subtle aspects of individual mutations.

Looking forward, our method is well-poised to investigate many outstanding questions in GPCR and drug receptor biology. First, individual GPCRs signal through multiple pathways, including pathways mediated by various G proteins and arrestins (*Galandrin et al., 2007*; *DeWire et al., 2007*; *Hilger et al., 2018*; *Luttrell, 2008*). We have only measured cAMP signaling in this manuscript, the primary signaling pathway of Gs-coupled GPCRs, but transcriptional reporters exist for the other signaling modalities and are compatible with our multiplexed approach. By leveraging transcriptional reporters for each of these pathways, we can understand the mechanisms that underpin signal transduction and biased signaling (*Reiter et al., 2012*). Second, GPCRs are often targeted by synthetic molecules with either unknown or predicted binding sites, and often have no known structures. We find ligands imprint a mutational signature on their receptor contacts which could potentially reveal the binding site for allosteric ligands. However, it should be noted that variation in receptor response to chemically diverse ligands at the cell surface may not reflect differences in downstream signal (*Tsvetanova et al., 2017*). We also found several regions on the external surface of the receptor where activating mutants are clustered. Since perturbations at these sites appear to increase receptor activity, they could potentially be targeted by positive allosteric modulators or allosteric agonists (*Thal et al., 2018*). Third, the identification of mutations that can stabilize specific conformations or increase receptor expression can aid in GPCR structure determination (*Serrano-Vega et al., 2008*; *Tate and Schertler, 2009*). Fourth, the development of stable cell libraries expressing human medicinally related GPCR variants can be combined with large-scale profiling against small molecule libraries to build very large-scale empirical maps for how small molecules modulate this broad class of receptors (*Botvinik and Rossner, 2012*; *Galinski et al., 2018*; *Jones et al., 2019*). Finally, our approach is generalizable to many classes of drug receptors where transcriptional reporters exist or can be developed (*O'Connell et al., 2016*), enabling the functional profiling, structural characterization, and pharmacogenomic analysis for most major drug target classes.

# Materials and methods

## Key resources table

| Reagent type (species) or resource | Designation | Source or reference | Identifiers | Additional information |
|---|---|---|---|---|
| Cell line (*Homo-sapiens*) | HEK293T | ATCC | CRL-3216 | |
| Cell line (*Homo-sapiens*) | HEK293TΔADRB2 + Landing Pad | This paper | | Construction Information found in Endogenous ADRB2 Deletion using CRISPR/Cas9 and Landing Pad Genome Editing Sections |
| Gene (*Homo-sapiens*) | ADRB2 | NCBI | Gene ID 154 | |
| Chemical compound, drug | Isoproterenol | Millipore Sigma | I5627 | |
| Chemical compound, drug | Forskolin | Millipore Sigma | F6886 | |
| Commercial assay or kit | Dual Glo Luciferase Assay | Promega | E2920 | |
| Recombinant DNA reagent | TALEN plasmids | Addgene | #51554 #51555 | |
| Recombinant DNA reagent | SpCas9 plasmid | Addgene | pX339 | |
| Sequence-based reagent | Oligonucleotide Microarray | Agilent | Custom Synthesis | |
| Commercial assay or kit | Nextseq Mid Output 300 cycle | Illumina | 20024905 | |
| Commercial assay or kit | Nextseq High Output 75 cycle | Illumina | 20024906 | |
| Strain, strain background (*Escherichia coli*) | Dh5 alpha | New England Biolabs | C2989K | |
| Antibody | AlexaFluor 488 Anti-Flag rat monoclonal | Thermo Fisher | MA1-142-A488 | (1:100) |
| Transfected construct (*Homo-sapiens*) | ADRB2 barcoded variant-reporter library | This paper | | Reagent Construction Information found in Variant Library Generation and Cloning Section |
| Commercial assay or kit | RNEasy Miniprep Kit | Qiagen | 74104 | |
| Commercial assay or kit | Plasmid Plus DNA Maxi Kit | Qiagen | 12963 | |
| Commercial assay or kit | Superscript IV | Thermo Fisher | 18091050 | |
| Commercial assay or kit | Lipofectamine 3000 | Thermo Fisher | L3000001 | |

*Continued on next page*

*Continued*

| Reagent type (species) or resource | Designation | Source or reference | Identifiers | Additional information |
|---|---|---|---|---|
| Commercial assay or kit | D1000 DNA Screen Tape | Agilent | 5067–5582 | |
| Commercial assay or kit | D1000 Reafents | Agilent | 5067–5583 | |
| Commercial assay or kit | SYBR FAST QPCR Master Mix | Roche | 07959362001 | |
| Commercial assay or kit | Zymo Clean Gel D NA Recovery Kit | Zymo Research | D4007 | |
| Commercial assay or kit | Zymo DNA Clean and Concentrator Kit | Zymo Research | D4013 | |
| Chemical compound, drug | CD293 | Thermo Fisher Scientific | 11913019 | |
| Software, algorithm | BBTools | Brian Bushnell | https://jgi.doe.gov/data-and-tools/bbtools/ | |
| Software, algorithm | Jensen-Shannon Conservation | https://doi.org/10.1093/bioinformatics/btm270 | | |
| Software, algorithm | OMA Orthology Database | https://doi.org/10.1093/nar/gkx1019 | | |
| Software, algorithm | FreeSASA | 10.12688/f1000research.7931.1 | | |
| Software, algorithm | EVmutation | doi:10.1038/nbt.3769 | | |
| Software, algorithm | Parasail | http://dx.doi.org/10.1186/s12859-016-0930-z | | |

## Cell line statement

We used HEK293T cells purchased from ATCC that were genetically modified in house. The identity of the lines have been verified with STR profiling and have tested negative for mycoplasma contamination.

## Experimental methods

### Endogenous ADRB2 deletion using CRISPR/Cas9

Cas9 and sgRNAs targeting the sole exon of ADRB2 were cloned (addgene: pX330) and transfected into HEK293T cells according to the protocol outlined in *Supplementary file 4*; *Ran et al., 2013*. After transfection, cells were seeded in a 96-well plate at a density of 0.5 cells/well. Wells were examined for single colonies after 3 days and expanded to 24-well plates after 7 days. Clones were screened for ADRB2 deletion by screening them for the inability to endogenously activate a cAMP genetic reporter when stimulated with the ADRB2 agonist isoproterenol. Clones were seeded side by side wild type HEK293T cells at a density of 7300 cells/well in a poly-D lysine coated 96-well plate. 24 hr later, cells were transfected with 10 ng/well of a plasmid encoding luciferase driven by a cyclic AMP response element and 5 ng/well of a plasmid encoding Renilla luciferase with lipofect-amine 2000. 24 hr later, media was removed and cells were stimulated with 25 μl of a range of 0 to 10 μM isoproterenol (Sigma-Aldrich) in CD293 (Thermo Fisher Scientific) for 4 hr. After agonist stimulation, the Dual-Glo Luciferase Assay kit was administered according to the manufacturer's instructions. Luminescence was measured using the M1000 plate reader (Tecan). All luminescence values

were normalized to Renilla luciferase activity to control for transfection efficiency in a given well. Data were analyzed with Microsoft Excel and R.

## Landing pad genome editing

The H11 locus was edited using TALEN plasmids received from Addgene (#51554, #51555). HEK293T cells were seeded at a density of 75 k cells in a 24-well plate. 24 hr after seeding cells were transfected with 50 ng LT plasmid, 50 ng RT plasmid, and 400 ng of the Linearized Landing Pad using Lipofectamine 2000. 2 days after transfection, cells were expanded to a six-well plate and one day after expansion 500 µg/ml hygromycin B (Thermo FIsher Scientific) was added to the media. Cells were grown under selection for 10 days. After selection, cells were seeded in a 96-well plate at a density of 0.5 cells/well. Wells were examined for single colonies after 3 days and expanded to 24-well plates after 7 days. gDNA was purified using the Quick-gDNA Miniprep kit (Zymo Research) from the colonies and PCR was performed with Hifi Master Mix to ensure the landing pad was present at the correct locus (LP001F and R). The reaction and cycling conditions are optimized as follows: 95°C for 3 min, 35 cycles of 98°C for 20 s, 63°C for 15 s, and 72°C for 40 s, followed by an extension of 72°C for 2 min. To ensure a single landing pad was present per cell, HEK293T cell lines with both singly and doubly integrated landing pads along with untransduced (WT) HEK293T cells were plated at $4 \times 10^5$ cells per 6-well. All landing pad cells were transfected the next day with 1.094 µg of both an attB-containing eGFP and mCherry donor plasmid and 0.3125 µg of the Bxb1 expression vector or a pUC19 control. Two singly integrated landing pad cell samples were also transfected with 2.1875 µg of either an attB-containing eGFP and mCherry donor plasmid with 0.3125 µg of the Bxb1 expression vector. Cells were transfected at a 1:1.5 DNA:Lipofectamine ratio with Lipofectamine 3000. 2 days later cells were passaged at 1:10 and were analyzed using flow cytometry 10 days later after four total passages. Samples were flown using the LSRII at the UCLA Eli and Edythe Broad Center of Regenerative Medicine and Stem Cell Research Flow Cytometry Core. Cytometer settings were adjusted to the settings: FSC – 183 V, SSC – 227 V, PE-Texas Red – 336 V, Alexa Fluor 488–275 V.

## Individual donor Bxb1 recombinase plasmid integrations

HEK293T-derived cells engineered to contain the Bxb1 Recombinase site at the H11 locus were seeded at a density of 350 k cells in a six-well plate (Corning). 24 hr after seeding cells were transfected with 2 µg Donor plasmid and 500 ng plasmid encoding the Bxb1 recombinase using Lipofectamine 3000 (Thermo Fisher Scientific). 3 days after transfection cells were expanded to a T-75 flask (Corning) and 8 µg/ml blasticidin (Thermo Fisher Scientific) was added one day after expansion. Cells were kept under selection 7–10 days and passaged twice 1:10 to ensure removal of transient plasmid DNA.

## Ligand-receptor activation luciferase assay for genomically integrated receptor/reporter constructs

HEK293T and HEK293T derived cells integrated with the combined receptor/reporter plasmids were plated at a density of 7300 cells/well in 100 uL DMEM in poly-D-lysine coated 96-well plates. 48 hr later, media was removed and cells were stimulated with 25 µl of a range of isoproterenol concentrations in CD293 for 4 hr. After agonist stimulation, the Dual-Glo Luciferase Assay kit was administered according to the manufacturer's instructions. Luminescence was measured using the M1000 plate reader. Data were analyzed with Microsoft Excel and R.

## Ligand-receptor activation q-RT PCR assay for genomically integrated receptor/reporter constructs

HEK293T and HEK293T-derived cells integrated with the combined receptor/reporter plasmids were plated at a density of 200 k cells/well in 2 mL DMEM in 6-well plates. 48 hr after seeding, media was removed and cells were induced with various concentrations of either forskolin (Sigma-Aldrich) or isoproterenol diluted in 1 ml of OptiMEM (Thermo Fisher) per plate for 3 hr. After stimulation, media was removed and 600 µL of RLT buffer (Qiagen) was added to each well to lyse cells. Lysate from each sample were homogenized with the QIAshredder kit (Qiagen) and total RNA was prepared from each sample using the RNeasy Mini Kit with the optional on-column DNAse step (Qiagen). Of

the total RNA per sample, 5 μg was reverse transcribed with Superscript III (Thermo-Fisher) using a gene-specific primer for the reporter gene and GAPDH (*Supplementary file 4*) according to the manufacturer's protocol. The reaction conditions are as follows: Annealing: [65°C for 5 min, 0°C for 1 min] Extension: [52°C for 60 min, 70°C for 15 min]. 10% of the RT reaction was amplified in triplicate for both genes, the reporter gene and GAPDH (*Supplementary file 4*), using the SYBR FAST qPCR Master mix (Kapa Biosystems) with a CFX Connect Thermocycler (Biorad). The reaction and cycling conditions are optimized as follows: 95°C for 3 min, 40 cycles of 95°C for 3 s and 60°C for 20 s. Reporter gene expression was normalized to GAPDH expression for each sample. Data were analyzed with Microsoft Excel and R.

## Variant library generation and cloning

The ADRB2 missense variant library was created by splitting the protein coding sequence into eight distinct segments (~52 a.a. each) and synthesizing all single amino acid substitutions for each segment separately as an oligonucleotide library (Agilent). 500 pg of the oligonucleotide library was amplified with biotinylated primers unique for each segment (*Supplementary file 4*) with the Real-Time Library Amplification Kit (Kapa Biosystems) on a CFX Connect Thermocycler (Biorad). The reaction and cycling conditions are as follows: 98°C for 45 s, X cycles of 98°C for 15 s, 65°C for 30 s, and 72°C for 30 s, followed by an extension of 72°C for 1 min. The number of cycles for the amplification was determined to ensure the amplification was in the exponential phase at least two cycles before the amplification reached saturation. The PCR products were cleaned up with the DNA Clean and Concentrator Kit (Zymo Research) and digested with restriction enzymes BamHI and BspQI, BbsI and BspQI, or BbsI and NheI (New England Biolabs). Digestions were cleaned up with the DNA Clean and Concentrator Kit and digested ends of the amplified library were removed by performing a streptavidin bead cleanup with the Dynabeads M-280 and the DynaMag (Thermo Fisher). Each library segment was to be cloned into a different vector that includes components of the ADRB2 reporter and the wild-type sequence portion of ADRB2 upstream of the segment being cloned. These eight different base vectors were digested (20 μg each) with restriction enzymes BamHI and BspQI, BbsI and BspQI, or BbsI and NheI. The base vectors were cleaned up with the DNA Clean and Concentrator Kit and the library segments were ligated into the base vectors (2.25 μg of vector with a 3:1 molar ratio of vector:insert, 900 μl reactions) with T4 DNA ligase (2,000,000 units/μl, New England Biolabs). The ligations were cleaned up with the DNA clean and Concentrator Kit and eluted into 25 μl. The purified ligations were placed on a 0.22 micron membrane filter (Millipore Sigma) floating in water in a 10-cm petri dish and dialyzed for 1 hr to remove excess salts that inhibit transformation. The ligations were then transformed into 5-alpha Electrocompetent cells (2 μl of ligation per bacterial aliquot, roughly five transformations per segment, New England Biolabs) directly into liquid culture. Cultures were grown at 30°C overnight to maintain library diversity and dilutions were plated on agarose plates to ensure transformation efficiency was high enough to cover the entire library (>100 transformants per library member). DNA was prepared 16 hr later with the DNA Miniprep Kit (Qiagen) and 20 mL of culture was prepped per segment. The vectors were digested (20 μg each) with BspQI and AgeI or NheI and AgeI (Qiagen). Vectors containing unique sequences corresponding to each library segment that complete the ADRB2 protein sequence and reporter were digested with the same restriction enzymes. These fragments were gel isolated from a 1% agarose gel using the Zymoclean Gel DNA Recovery Kit (Zymo Research). These secondary fragments were cloned into the library vectors with the same protocol as the previous cloning step. DNA was prepared 16 hr later with the Plasmid Plus DNA Maxiprep Kit (two maxipreps of 100 mL culture each per library, Qiagen).

## Variant-barcode mapping

After the initial cloning of the variant fragments from the oligonucleotide library into each segment's corresponding base vector, the random barcode attached to each variant was associated to its variant with paired-end sequencing. Each plasmid was amplified with two rounds of PCRs with distinct primer sets for each segment (*Supplementary file 4*) with HiFi DNA Master Mix (Kapa Biosystems). For the first round of amplification, the reaction and cycling conditions were optimized as follows: 98°C for 30 s, 10 cycles of 98°C for 8 s, 64°C for 15 s, and 72°C for 10 s, followed by an extension of 72°C for 2 min. These amplicons were gel isolated from a 1% agarose gel using the Zymoclean Gel

DNA Recovery Kit. Prior to the second round of amplification, the number of cycles to amplify was determined by performing qPCR with the SYBR FAST QPCR Master Mix (Kapa) on the CFX Connect Thermocycler according to the manufacturer's instructions. The Cq determined from the QPCR plus an additional two cycles was used as the number of cycles to amplify the libraries for the second round of amplification. For the second round of amplification, the reaction and cycling conditions were optimized as follows: 98℃ for 30 s, X cycles of 98℃ for 8 s, 62℃ for 15 s, and 72℃ for 10 s, followed by an extension of 72℃ for 2 min. These amplicons were gel isolated from a 1% agarose gel using the Zymoclean Gel DNA Recovery Kit. Kit. Library concentrations were quantified using a TapeStation 2200 (Agilent) and a Qubit (Thermo Fisher). The libraries were sequenced with paired end 150 bp reads on a NextSeq 500 in medium-output mode and paired end 250 bp reads on a MiSeq (Illumina).

## Variant library Bxb1 recombinase plasmid integrations

HEK293T-derived cells engineered to contain the Bxb1 recombinase site at the H11 locus and deletion of endogenous ADRB2 were seeded at a density of 2.13 million cells per dish in 6 100 mm x 20 mm tissue-culture treated culture dishes (Corning). 24 hr after seeding cells were transfected with 11.5 ug Donor plasmid and 2.9 μg plasmid encoding the Bxb1 recombinase using Lipofectamine 3000. Three days after transfection, cells were expanded to T-225 flasks (Corning) and 8 μg/ml blasticidin was added 1 day after expansion. Cells were kept under selection 7–10 days and passaged 1:10 four times to ensure removal of transient plasmid DNA.

## Multiplexed variant functional assay agonist stimulation, RNA preparation and sequencing

HEK293T-derived cells engineered to contain the Bxb1 recombinase site at the H11 locus, deletion of endogenous ADRB2, and integration of the ADRB2 mutagenic library were seeded at a density of 3,200,000 cells per dish in 150 mm x 25 mm tissue-culture treated culture dishes. 10 dishes were seeded for each biological replicate of each drug condition. 48 hr after seeding, media was removed and cells were induced with various concentrations of either forskolin or isoproterenol diluted in 9 ml of OptiMEM per plate for 3 hr. After stimulation, media was removed and 3.24 ml of RLT buffer was added to each well to lyse cells. Lysate from dishes belonging to the same replicate were pooled and vortexed thoroughly. 5 ml of lysate from each sample were homogenized with the QIAshredder kit and total RNA was prepared from each sample using the RNeasy Midi Kit with the optional on-column DNAse step (Qiagen) and eluted into 500 μl H$_2$O. 40 reverse transcriptase reactions were carried out for each sample using the Superscript IV RT kit (Thermo Fisher). For each reaction 11 ul of total RNA were added to 1 μl dNTPs (Qiagen) and 1 μl 2 uM RT primer (*Supplementary file 4*). The primers were annealed to the template by heating to 65℃ for 5 min and cooling down to 0℃ for 1 min. After annealing, 4 μl of RT buffer, 1 μl DTT, 1 μl of RNAseOUT, and 1 μl SSIV were added to the mixture and cDNA synthesis was performed. The reaction and cycling conditions are as follows: 52℃ for 1 hr, 80℃ for 10 min. cDNA from the same sample was pooled together and treated with 100 ug/ml RNAse A (Thermo Fisher) and 200 U of RNase H (Enzymatics) at 37℃ for 30 min. cDNA was concentrated using the Amicon Ultra 0.5 mL 30 k Centrifugal Filter (Millipore) according to the manufacturer's instructions with a final spin step time of 15 min. To determine the number of cycles necessary for library amplification in preparation for RNA-seq, 1 μl of cDNA from each sample was amplified with SYBR FAST QPCR Master Mix according to the manufacturer's instructions using primers for library amplification and adaptor addition (*Supplementary file 4*). Each sample was subsequently amplified for four cycles more than the Cq calculated in the QPCR run adjusting for sample volume. The entire volume of concentrated cDNA for each sample was amplified with sequencing adaptors using NEB-Next High-Fidelity 2x PCR Master Mix (New England Biolabs): 25 μl Master Mix, 2.5 μl of both 10 uM forward and reverse primer (*Supplementary file 4*), 4 μl of cDNA, and 16 μl H$_2$O. The reaction and cycling conditions are as follows: 98℃ for 30 s, X cycles of 98℃ for 8 s, 66℃ for 20 s, and 72℃ for 10 s, followed by an extension of 72℃ for 2 min. Amplified DNA was purified with the DNA Clean and Concentrator kit and gel isolated from a 1% agarose gel with the Zymoclean Gel DNA Recovery Kit. Library concentrations were quantified using a TapeStation 2200 and a Qubit. The libraries were sequenced with an i7 index read and a single end 75 bp read on a

NextSeq 500 in high-output mode. The coverage for the various experimental conditions are as follows:

| Condition | Repeat | Reads |
|---|---|---|
| 0 | 1 | 46811302 |
| 0 | 2 | 43527478 |
| 0.150 | 1 | 51795485 |
| 0.150 | 2 | 47528508 |
| 0.625 | 1 | 45295157 |
| 0.625 | 2 | 58560000 |
| 5 | 1 | 48206666 |
| 5 | 2 | 34977852 |
| F | 1 | 51172562 |
| F | 2 | 42013807 |
| F_5 | 1 | 41727633 |
| F_5 | 2 | 38259270 |

## Immunostaining and flow cytometry for surface expression

β2AR variants were cloned into the mammalian expression vector pCI with an N-terminal FLAG tag. HEK293T cells were seeded in a 96-well tissue-culture-treated plate (Genesee Scientific) at 30,000 cells/well. 24 hr after seeding 50 ng of each receptor variant and 50 ng of carrier DNA (pUC19) was transfected in triplicate per variant per well with Lipofectamine 3000 (Thermo Fisher Scientific) according to the manufacturer's protocol. Negative control wells were transfected with 100 ng of carrier DNA. 48 hr after transfection, media was aspirated and each well was washed with 100 µl of Cell Dissociation Buffer, enzyme free, PBS (Thermo Fisher Scientific). 100 µl of Cell Dissociation Buffer was added to each well and plates were left at room temperature for 20 min. 100 µl of PBS +0.5% FBS was added to each well and mixed. 200 mcL of cell slurry was transferred to a U-bottom 96-well plate (NEST Scientific) and plates were spun down at 488 x g for 5 min. Supernatant was removed with a medium strength flick and cells were resuspended in AlexaFluor 488 conjugated monoclonal Anti-FLAG (diluted 1:100 v/v in PBS+0.5% FBS; Thermo Fisher Scientific). Plates were covered with foil and incubated at 4C for 30 min. After incubation, 150 µl of flow buffer was added to each well, mixed, and plates were centrifuged at 488 x g for 5 min. Supernatant was removed with a medium strength flick and cells were resuspended in 200 ul PBS+0.5% FBS. Cells were analyzed with a MACS Quant 10 Flow Cytometer. Cytometer settings were adjusted to the settings: FSC – 385 V, SSC – 375 V, Alexa Fluor 488–375 V. Data was analyzed using FlowJo. First, singlets were gated on FSC-A vs. SSC-A and SSC-A vs. SSC-H. AlexaFluor fluorescence was initially gated on the no receptor negative control and the geometric mean of this gate was used as the measurement of surface expression. Data was plotted using R.

## Quantification and statistical analysis

### Barcode mapping

We used the BBTools suite (https://jgi.doe.gov/data-and-tools/bbtools/) of programs to process our sequencing data using the default settings unless otherwise noted. First, we used BBDuk2 to filter out any reads matching PhiX (k = 23, mink = 11, hdist = 1) and to trim off any Illumina sequencing adapters. We then used BBMerge to merge our paired end reads. We performed another round of trimming with BBDuk2 to ensure no adapters were left over after merging and to remove any sequence with an N base call. After merging and trimming the reads, we used a custom Python script (bcmap.py) to generate a consensus nucleotide sequence for each barcode.

Briefly the script works as follows. First, we split each read into the 15 nt barcode and its corresponding variant. We then generate a dictionary that maps each barcode to its list of unique sequences and their counts. To enable majority basecalls, we drop any barcode that has less than three

reads. We then pass the barcodes through a series of filters to eliminate potential errors introduced by barcodes that are mapped to multiple variants. Since we barcoded and mutagenized the ADRB2 gene in separate pieces, barcodes can be contaminated with variants from different parts of the ADRB2 gene. We address this case by using BBMap to align every barcode's sequences to the ADRB2 reference and consider that barcode to be contaminated if any sequence aligns >5 nt away from the most common sequence. Another source of contamination comes from the chip-synthesized library itself, which contains a significant number of single base deletions. We consider a barcode contaminated if it has any sequences of different lengths as it is unlikely that a single base deletion will come from an Illumina sequencer by chance. However, these filters would not catch the case where a barcode is contaminated with variants from the same piece of ADRB2. As we only synthesized the missense variants, we expect variants within the same piece of ADRB2 to be a Levenshtein distance of 4 from each other on average (approximately two changes to WT and two changes to a new codon). Thus, we drop any barcode that has a sequence with >1 read at a Levenshtein distance of 4 away from that barcode's most common sequence. Lastly, we generate a consensus sequence by taking the majority base call at each position and call an N at any ties.

After we associate each barcode with its consensus sequence, we use a series of different alignments to determine that sequence's identity. To find the designed missense variants in our library, we use BBMap to search for barcodes that have an exact alignment to them. To find frameshift mutations, we use BBMap to align the consensus sequences to the ADRB2 reference and parse the resulting CIGAR strings for indels with a simple python script (classify-negs.py). Finding synonymous mutants required more processing as each sub-library did not start at a complete codon. We first used the rough BBMap alignment to determine what ADRB2 chunk each sequence was associated with. We then used a custom python script (synon-filter.py) to trim up to the last whole clonal codon, as the first few codons of each sequence were part of the clonal backbone and are unlikely to have any errors. Finally, we translated the resulting sequences, aligned the protein sequence to the ADRB2 coding sequence with a Smith-Waterman aligner from the Parasail library (*Daily, 2016*) (https://github.com/jeffdaily/parasail, copy archived at swh:1:rev:2fee307b6209d4a26be144f3e008-de0e02e1c8db), and retained perfect translations with the correct length.

## Data normalization

We incubated our cellular library with forskolin to activate the cAMP reporter in each cell, providing an agonist-independent measurement of maximal reporter activity. This measurement can be used to approximate cellular copy number. To ensure that barcodes with low cellular representation are excluded from our analyses, we require all barcodes to be present in both forskolin repeats, and filter out any barcodes with a mean reads per million less than 0.2 (~8–10 reads at our sequencing depth). We also excluded barcodes with high forskolin counts (>=10 RPM) as they are systematically less induced in the drug conditions relative to other barcodes. Next, we require that all of the barcodes in the forskolin condition are also present in our drug conditions, and set any missing barcodes to 0. We then add a pseudocount that is scaled relative to the condition with the fewest number of reads (N/min(N)), and normalize each condition to its read depth (including added pseudocounts) (*Bloom, 2015*). Finally, we divide this value by its associated forskolin value to control for variation in cellular abundance.

Since each variant in our library was associated with a median of 10 barcodes, we took the average of all barcodes. We then defined activity as the ratio of these values to the value of the mean frameshift. Finally, we averaged the relative activities of our two repeats together and used propagation of uncertainty to combine their standard deviations.

## Conservation, EVMutation, and gnomAD

To calculate sequence conservation at a species level, we aligned 55 ADRB2 orthologs from the OMA database (entry: HUMAN24043) using MAFFT with the default settings (mafft `-reorder -auto`). For Class-A GPCRs, we retrieved the multiple sequence alignment from GPCRdb. We then used the Jensen-Shannon Divergence (*Altenhoff et al., 2018*; *Capra and Singh, 2007*; *Hopf et al., 2017*) to score both these alignments. We only considered conservation scores at positions in the MSA that contained residues from the β2AR. For both EVMutation and gnomAD, we simply downloaded the results for ADRB2. We considered residues in gnomAD to be potentially loss of function

if their mean activity plus the standard error of the mean was less than two standard deviations from the mean of the frameshift distribution.

## Unsupervised learning

We performed a number of preprocessing steps before running UMAP on our data. First, we grouped amino acids into eight different classes based on their physicochemical properties ((+) - R, H, K; (-) - D, E; Aromatic - F, W, Y; Amide - N, Q; Nucleophilic - C, S, T; Hydrophobic - I, L, V, M; Small - G, A; Proline - P) and averaged their relative activities. Next, we standardized the log2 relative activity values of each group and used mean imputation to model missing data for any missing AA groups at a given position. Finally, we combined the data from every drug condition into a 412 × 32 design matrix in which the columns are an AA group at a specific condition and the rows are the positions in the protein.

With our data processed, we used the R implementation of UMAP to run hyperparameter search (https://github.com/jlmelville/uwot; copy archived at swh:1:rev:b449908402ba0ab5348c22cd2620e-fe23de01012; *Melville, 2019*) of all combinations of UMAP embeddings with the parameters n_neighbors = (4, 8, 16, 32) and n_components = (2, 3, 4, 5, 6, 7, 8, 9, 10), holding min_dist = 0 and n_epochs = 2000 constant. This provided a variety of different representations of our data that we used HDBSCAN (*Campello et al., 2013*) to search for clusters in these embeddings (R package dbscan; minPts = 10). To ease interpretation of the clustering, we plotted the HDBSCAN results onto a 2D UMAP embedding with the following parameters: n_neighbors = 4, min_dist = 0, n_components = 2, n_epochs = 2000, and random_state = 3308004 using the Python implementation (*McInnes and Healy, 2018*) (https://github.com/lmcinnes/umap; copy archived at swh:1:rev:2b9a2521b4c6d5f084278b2e967040e3020dac9d). We found the cluster assignments to be largely robust across the different embeddings, and used them to guide our manual cluster assignment.

## Structural modeling and solvent accessible surface area

Molecular graphics and analyses were performed with UCSF Chimera (*Pettersen et al., 2004*) and PyMol. To determine if a given position in the $\beta_2$AR points into the core of the protein or into the lipid membrane, we used FreeSASA (*Mitternacht, 2016*) (version 2.0.3) to calculate the Solvent Accessible Surface Area (SASA) of the $G_s$-bound $\beta_2$AR (PDB: 3SN6). The $G_s$ occludes the intracellular surface of the $\beta_2$AR thereby reducing the SASA of residues on the intracellular surface. Similarly, the extracellular surface is mostly blocked by the extracellular loops. Finally, we used the Orientations of Proteins in Membranes (OPM) database (*Lomize et al., 2012*) to filter out any residues outside of the lipid membrane from our analyses. To quantify charge sensitivity, we calculated the average activity for H, K, R, D, and E substitutions at each agonist concentration for residues in the lipid membrane. We then multiplied the values by −1 and standardized the results within each agonist concentration group such that the values were mean-centered and scaled by their standard deviation. We calculated hydrophobic sensitivity (I, L, V, M) in an analogous manner. Next, we classified residues that had above average charge sensitivity and below average hydrophobic sensitivity as being exclusively charge sensitive. Conversely, we classified residues that had above average charge sensitivity and above average hydrophobic sensitivity as being intolerant.

## Structural and sequence analysis of Class A GPCRs

For the structural analysis, the crystal structures of class A GPCRs were obtained from PDB (*Berman et al., 2000*). In order to compare structures across the different sub-families of class A GPCRs, structures of representative examples from each subfamily were chosen. In order to compare structures across the conformational states in different GPCRs, the structures of beta-2 adrenergic receptor, M2 muscarinic receptor, kappa-opioid receptor, and mu-opioid receptor were chosen. A2A receptor and mammalian rhodopsin were excluded as they lacked a Trp residue in the first extracellular loop. These receptors were chosen due to the availability of pairs of inactive state and active state structures. The inactive state structures were identified based on the presence of co-crystallized antagonist/inverse-agonist and the active state structures were identified based on the presence of a co-crystallized agonist and co-complexed interacting partner at the G-protein-coupling site. Structural alignment and measurement of inter-atomic distances were performed using

PyMOL (https://pymol.org). The structure alignment was performed over the sequence stretch between the conserved Trp/Phe residue in the first extracellular loop and the canonical disulfide bridge forming Cys3 × 25 (GPCRdb number) present on TM3.

For the sequence analysis, the sequence alignment of Class A GPCRs was obtained from GPCRdb (*Pándy-Szekeres et al., 2018*). The alignment was filtered for receptors that contained the canonical disulfide bridge forming on TM3 residue in ECL1, which gave a total of 202 GPCRs sequences. Using this alignment, the sequence logo was made using the Weblogo program (https://weblogo.berkeley.edu/logo.cgi).

## Statistical tests
All statistical tests unless otherwise noted are the two-sided Mann-Whitney U test and were performed in R (version 3.5.x) using the wilcox.test function.

## Software
All codes are available at (https://www.github.com/KosuriLab/b2-dms; *Jones, 2020* copy archived at swh:1:dir:09b4931491e1c9f9ee2c90c5687f44efa6464373). Sequencing data can be accessed from the sequencing read archive (SRA) with the accession number SRP247450 or from the Gene Expression Omnibus (GEO) with the accession number GSE144819. To avoid potential visual distortions in the heatmap, we used perceptually uniform color maps (*Crameri, 2018*). For parallelization, we employed GNU Parallel (*Tange O, 2011*).

# Acknowledgements
We thank the Kosuri Lab for helpful discussions, the UCLA Broad Stem Cell Research Center Sequencing and Flow Cytometry Core, and the Technology Center for Genomics and Bioinformatics for providing next-generation sequencing. We thank Deborah Marks and Jung-Eun Shin for advice implementing EVmutation. We thank Robert J Lefkowitz and Laura Wingler for suggesting control mutations to test and technical guidance. We thank Hiroaki Matsunami, Joshua S Bloom, Rishi Jajoo and Rocky O Cheung for expert technical assistance. Funding: National Science Foundation, Brain Initiative (1556207 to SK), Ruth L Kirschstein National Research Service Award (GM007185 to NBL), the USPHS National Research Service Award (5T32GM008496 to EMJ), the NIH (DP2GM114829 to SK and R01 GM127359 to R.O.D), the Medical Research Council (MC_U105185859 to MMB and AJV), and UCLA. Data and materials availability: Processed data and analysis scripts are available on https://github.com/KosuriLab/b2-dms. Raw data is available with accession number XXXXXX. Plasmids and cell lines are available upon request.

# Additional information

### Competing interests
Eric M Jones: holds equity and is employed by Octant, Inc, a company to which patent rights based on this work have been licensed (Application No. 62/528,833). Nathan B Lubock: employed by and holds equity in Octant Inc to which patent rights based on this work have been licensed (Application No. 62/528,833). Sriram Kosuri: holds equity and is employed by Octant, Inc, a company to which patent rights based on this work have been licensed to (Application No. 62/528,833). The other authors declare that no competing interests exist.

### Funding

| Funder | Grant reference number | Author |
| --- | --- | --- |
| National Science Foundation | 1556207 | Sriram Kosuri |
| National Institutes of Health | GM007185 | Nathan B Lubock |
| National Institutes of Health | 5T32GM008496 | Eric M Jones |
| National Institutes of Health | DP2GM114829 | Sriram Kosuri |
| Medical Research Council | MC_U105185859 | AJ Venkatakrishnan |

|  |  | M Madan Babu |  |
|---|---|---|---|
| National Institutes of Health | GM127359 | Ron O Dror |  |

The funders had no role in study design, data collection and interpretation, or the decision to submit the work for publication.

## Author contributions

Eric M Jones, Conceptualization, Formal analysis, Supervision, Validation, Investigation, Visualization, Methodology, Writing - original draft, Project administration, Writing - review and editing; Nathan B Lubock, Data curation, Software, Formal analysis, Visualization, Writing - original draft, Writing - review and editing; AJ Venkatakrishnan, Formal analysis, Investigation, Visualization, Writing - original draft, Writing - review and editing; Jeffrey Wang, Daniel Cancilla, Megan Satyadi, Validation, Investigation; Alex M Tseng, Joseph M Paggi, Naomi R Latorraca, Formal analysis, Writing - review and editing; Jessica E Davis, Validation; M Madan Babu, Ron O Dror, Formal analysis, Writing - original draft, Writing - review and editing; Sriram Kosuri, Conceptualization, Data curation, Formal analysis, Supervision, Funding acquisition, Methodology, Writing - original draft, Project administration, Writing - review and editing

## Author ORCIDs

Eric M Jones (iD) https://orcid.org/0000-0002-6648-1965
Nathan B Lubock (iD) https://orcid.org/0000-0001-8064-2465
AJ Venkatakrishnan (iD) https://orcid.org/0000-0003-2819-3214
Sriram Kosuri (iD) https://orcid.org/0000-0002-4661-0600

## Decision letter and Author response

Decision letter https://doi.org/10.7554/eLife.54895.sa1
Author response https://doi.org/10.7554/eLife.54895.sa2

# Additional files

## Supplementary files

• Supplementary file 1. List of species for evolutionary analysis. A table describing the list of 55 species used for the analysis of evolutionary constraint of residues from the OMA database.

• Supplementary file 2. Processed data. A table with the processed data used in this study. Includes the position, mutation, min activity, max activity, average activity, propagated uncertainty, coefficient of variation, number of barcodes for that mutation, mutation class, and an annotation of where in the β2AR the position is.

• Supplementary file 3. Mutational tolerance. A table containing positions in the β2AR with their mutational tolerance, rank order, and an annotation for known positions in the receptor.

• Supplementary file 4. List of primers used in this study. A table documenting the DNA sequences and application of important primers in this study.

• Transparent reporting form

## Data availability

Sequencing data are available on GEO under the accession code GSE144819.

The following dataset was generated:

| Author(s) | Year | Dataset title | Dataset URL | Database and Identifier |
|---|---|---|---|---|
| Jones EM, Lubock NB, Venkatakrishnan A, Wang J, Tseng AM, Paggi JM, Latorraca NR, | 2020 | Structural and functional characterization of G protein-coupled receptors with deep mutational scanning | https://www.ncbi.nlm.nih.gov/geo/query/acc.cgi?acc=GSE144819 | NCBI Gene Expression Omnibus, GSE144819 |

Cancilla D, Satyadi M, Davis JE, Babu MM, Dror RO, Kosuri S

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
