## [Decision Letter]

**Acceptance summary:**

A mutation scanning procedure for amino acid replacements in G protein-coupled receptors is described and the outcome detected with cAMP-induced transcription of a luciferase reporter. Using the human beta-2 adrenergic receptor as proof of principle, the authors have investigated almost every possible amino acid replacement throughout the sequence. One interesting new observation is a conserved 'latch' involving three highly conserved residues in the receptor's first extracellular loop.

**Decision letter after peer review:**

Thank you for submitting your article "Structural and Functional Characterization of G Protein-Coupled Receptors with Deep Mutational Scanning" for consideration by *eLife*. Your article has been reviewed by three peer reviewers, and the evaluation has been overseen by a Reviewing Editor and Richard Aldrich as the Senior Editor. The following individuals involved in review of your submission have agreed to reveal their identity: James S Fraser (Reviewer #1); Aashish Manglik (Reviewer #2).

The reviewers have discussed the reviews with one another and the Reviewing Editor has drafted this decision to help you prepare a revised submission.

Summary:

All three reviewers and myself agree that this manuscript describes an interesting and potentially very useful scanning procedure of amino acid replacements in G protein-coupled receptors, exemplified by an impressively extensive analysis of the human beta-2 adrenergic receptor. Almost all possible mutations were evaluated after agonist stimulation, measured as cAMP-induced transcription of a luciferase reporter. The application of the procedure on beta-2 confirms several previous observations and thereby serves as proof of concept for this approach. It adds a few new observations, especially the proposed conserved 'latch' involving three highly conserved residues in EL1.

Essential revisions:

Detailed conclusions about mutation outcomes are limited by the fact that the one and only assay measures functional response and hence cannot distinguish mutational impact on all the preceding steps including biosynthesis, folding, intracellular transport, ligand binding, conformational change, G protein coupling and receptor internalization. Mutations that affect the functional output in an indirect fashion are likely to occur and this possibility should be discussed further. Ideally, it would be desirable that the authors could present quantification of cell surface expression for at least a subset of the deleterious mutants. Hopefully, such data has been collected.

Another aspect that is (as noted) important is the level of noise in the system. For example, it is not at all clear why it was necessary to use on average 10 barcodes per mutant, when other studies employing DNA-level abundance of barcodes have gotten away with smaller numbers of barcodes per variant.

Do all pairwise combinations exhibit equal reproducibility, or is it possible to model measurement error (e.g., making use of the number of read counts) as other studies have done to estimate error in individual measurements? Perhaps then we can be confident in some subset of residue-level measurements. Error estimates could then be propagated to higher-level aggregate summaries, e.g., average score at each given position and for the various missense variant types (hydrophobic, polar etc). Representative scatterplots between barcode replicates for a subset would be informative (ideally, with and without forskolin normalization). In addition, it would be interesting to see if sequence content of barcodes correlate with error estimates (e.g. certain barcode sequences might destabilize the transcripts, resulting in artificially lowered scores for a given mutation and vice versa). Error estimates could also be useful, e.g., in ranking the most intolerant amino acids, where ranking is based on the estimate at the more conservative end of a confidence interval. Also, with error estimates, statements like "we obtained measurements for 99.6% (7,800/7,828) of possible missense variants" could be replaced with statements like "we obtained reliable measurements for X% of possible missense variants.

Considering that the output assay as mediated by cAMP, the authors might want to common if the approach is limited to receptors coupling via G-alpha-s.

When the authors discuss which positions are conserved and which are not, it is not always clear whether they mean among ADRB2 orthologs or perhaps across adrenergic receptor subtypes or perhaps for the entire GPCR class A. Which receptors and species are compared? It is also essential to describe the range of species. The authors refer in a couple of places to 55 ADRB2 orthologs (Figure 2 legend, Figure 3—figure supplement 1 legend, subsection “Conservation, EVMutation, and gnomAD”) but do not specify which range of species was included in this data set. For instance, it makes a huge difference if it's mammals or vertebrates.

One of the major findings is the identification of the conserved EL2 motif WxxGxxxC, proposed to work as a 'latch'. It would have been very interesting indeed to see this hypothesis tested in some way, but hopefully this will come in the near future.

Likewise, the observations that distal mutations in the N-terminus and C-terminus lead to constitutive activity invites further studies. Is it possible to say something about this based upon mutagenesis already reported in the literature for beta-2 or other class A receptors? It appears likely that especially the N-terminal mutations may compromise biosynthesis and handling in ER and Golgi, why the caveat should be mentioned early in the manuscript that differences in expression level may explain the observed output results.

The comparison of the latch with two receptors that are closely related to each other (opioid kappa and mu) seems a bit superfluous. It would be more interesting if a few completely different peptide receptors were included in the comparison.

The observation of the EL1 latch has some precedence, see review by Hulme in TIPS, 2013, Figure 3A. Please check if this should be cited.

ADRB2 variants were synthesized in oligonucleotide microarrays split into 8 segments and integrated into the cell line. Additional details on the scheme and numbers/statistics on coverage, library wt representation, and evenness would be important to discuss and show – especially for reproducibility. (Rubin et al., Genome Biology, 2017).

The authors conduct the DMS experiment under four different isoproterenol conditions and normalize measurements to forskolin treatments. Experimental details on the forskolin activation in their assay or reference for this treatment would aid in interpreting the normalization approach.

What exactly distinguishes the globally intolerant clusters (clusters 1 and 2) in Figure 4? It seems there is a tighter range of activity to isoproterenol in cluster 2 than in 1 for all mutations and chemical properties, but does this get ranked differently than cluster 1?

---

## [Author Response]

Essential revisions:Detailed conclusions about mutation outcomes are limited by the fact that the one and only assay measures functional response and hence cannot distinguish mutational impact on all the preceding steps including biosynthesis, folding, intracellular transport, ligand binding, conformational change, G protein coupling and receptor internalization. Mutations that affect the functional output in an indirect fashion are likely to occur and this possibility should be discussed further. Ideally, it would be desirable that the authors could present quantification of cell surface expression for at least a subset of the deleterious mutants. Hopefully, such data has been collected.

We agree with the reviewer that our assay does not discriminate between mutations that affect signaling and surface expression. To address this concern, we cloned and measured the surface expression (FLAG-tag immunostaining and Flow Cytometry) of 11 mutants that of residues that we discuss in the manuscript. We compared these mutants to both the wild type receptor and three mutations previously described to have severely impaired expression.

We added the following to the main text:

“Additionally, we evaluated surface expression for a subset of W99^23x50^ and G102^3x21^ mutants (Figure 6—figure supplement 1B). Relative to three previously characterized mutants with severely impaired surface expression(Parmar et al., 2017) and wild type β2AR, the mutants exhibited mildly impaired to normal surface expression – supporting a role in signaling for these residues.”

We have included this figure in Figure 6—figure supplement 1 and added a detailed description of our surface expression protocol to the Materials and methods.

Another aspect that is (as noted) important is the level of noise in the system. For example, it is not at all clear why it was necessary to use on average 10 barcodes per mutant, when other studies employing DNA-level abundance of barcodes have gotten away with smaller numbers of barcodes per variant.

Each experimental system for a given DMS will have different intrinsic levels of variation between measurements due to a number of factors. For example, an assay run in *E. coli* can feasibly be run with billions of cells versus an assay in human cell lines where you are limited to fewer cells. This difference in cellular coverage per variant can contribute to differences in measurement variation between two such assays. For example, in Figure 1—figure supplement 1F, we show that correlation between repeats markedly improves as we increase the number of cells per barcode and physical amount of RNA in the RT reaction.

Aside from controlling for sequence specific barcode effects, barcodes for the same variant serve as replicates for variant measurements as the standard error goes by the inverse square root of the number of barcodes. This helps understand the noise in our assay, and gain power for calling variant activity.

Lastly, because we are measuring the transcription of these barcodes, the sequence content of individual barcodes can influence the expression of the transcript, and we can average this effect out by having many barcodes per variant. This is not an issue when measuring DNA barcode abundance.

We thank the reviewer for bringing this up and have added “Of note, we aimed for 10 barcodes per variant in order to account for any effects individual barcodes will have on reporter transcription and serve as statistical replicates for each variant.” to the manuscript.

Do all pairwise combinations exhibit equal reproducibility, or is it possible to model measurement error (e.g., making use of the number of read counts) as other studies have done to estimate error in individual measurements? Perhaps then we can be confident in some subset of residue-level measurements. Error estimates could then be propagated to higher-level aggregate summaries, e.g., average score at each given position and for the various missense variant types (hydrophobic, polar etc). Representative scatterplots between barcode replicates for a subset would be informative (ideally, with and without forskolin normalization).

Here we model measurement error by taking the mean and SD of the forskolin ratios of all barcodes associated with a mutation (see Materials and methods for details). We then average our two repeats together and use error propagation to combine the SDs. Scatter plots between replicates at the barcode level (reads per million) and at the variant level (mean forskolin ratio) for EC_100_ are shown in Figure 2—figure supplement 1A.

To provide another sense of the noise in our assay, we’ve plotted the distribution of the coefficient of variation (CV) for each mutant in our assay (see Author response image 1). Furthermore, these data (with error estimates) are provided in Supplementary file 2 for any party that would be interested in more sophisticated analyses. More broadly, we agree with the reviewers’ sentiment that further mining of these data could reveal additional insights into the structure-function relationship of the beta-2 adrenergic receptor.

In addition, it would be interesting to see if sequence content of barcodes correlate with error estimates (e.g. certain barcode sequences might destabilize the transcripts, resulting in artificially lowered scores for a given mutation and vice versa).

**Author response image 2. respfig2:** 

A cursory analysis suggests that there is no correlation between barcode sequence and error estimates. In Author response image 2 we are showing a representative plot (10,000 barcodes in one repeat of the 0.625 μm Isoproterenol condition) of standard score for each barcode versus the GC content of that barcode. Note the standard score here is (xi−μ)/σ, where xi is the forskolin ratio of that barcode, μ is the mean forskolin ratio of the mutant that barcode corresponds to, and σ is the standard deviation of the forskolin ratio of the mutant. Obviously this does not preclude there being an effect, and is one of the motivations for having multiple barcodes per variant.

Error estimates could also be useful, e.g., in ranking the most intolerant amino acids, where ranking is based on the estimate at the more conservative end of a confidence interval.

Per the reviewers’ suggestion, we’ve propagated the error for our mutational tolerance measurements (recall mutational tolerance is the average effect of all of the mutations at a given position). In Author response image 3 we’ve plotted Figure 5C with the positions ranked by mutational tolerance (as before) or by mutational tolerance + one standard deviation.

**Author response image 3. respfig3:** 

The two rankings appear quite similar visually. Indeed, correlation between the two rankings are almost perfect (Spearman’s rho = 0.995), especially amongst the top 15 mutants that we highlight in the text as shown in Author response image 4. Given the similarity, we’ve elected to keep Figure 5C as is.

**Author response image 4. respfig4:** 

Also, with error estimates, statements like "we obtained measurements for 99.6% (7,800/7,828) of possible missense variants" could be replaced with statements like "we obtained reliable measurements for X% of possible missense variants.

We will define our measurement to be reliable if it has a coefficient of variation < 1 (see above for the CV distributions). With this cutoff, we can reliably call between 95-99% (7,461-7,749 depending on the agonist concentration) of the 7,828 possible variants. We have amended the sentence to reflect this update (subsection “Measurement of mutant activities and comparison to evolutionary metrics”).

Considering that the output assay as mediated by cAMP, the authors might want to common if the approach is limited to receptors coupling via G-alpha-s.

While we have only reported and developed cAMP signaling for this approach thus far, we believe one of the strengths of this assay is the generalizability to other signaling outputs. For example, the NFAT genetic reporter is an equivalent way to measure calcium signaling for Gq-coupled receptors; indeed our preliminary data for a different receptor indicates that this works well. Additionally, Gi-coupled receptors signal by inhibiting cAMP, therefore the CRE genetic reporter described in this manuscript can be utilized by inverting the interpretation of the functional score. More broadly, transcriptional reporters exists for many protein classes, including nuclear hormone receptors, kinases, ion channels, transcription factors, and broad functionalities like proximity and localization assays.

For clarity, we added this sentence to the Discussion in the section addressing future directions:

“We have only measured cAMP signaling in this manuscript, the primary signaling pathway of Gs-coupled GPCRs, but transcriptional reporters exist for the other signaling modalities and are compatible with our multiplexed approach.”

When the authors discuss which positions are conserved and which are not, it is not always clear whether they mean among ADRB2 orthologs or perhaps across adrenergic receptor subtypes or perhaps for the entire GPCR class A. Which receptors and species are compared? It is also essential to describe the range of species. The authors refer in a couple of places to 55 ADRB2 orthologs (Figure 2 legend, Figure 3—figure supplement 1 legend, subsection “Conservation, EVMutation, and gnomAD”) but do not specify which range of species was included in this data set. For instance, it makes a huge difference if it's mammals or vertebrates.

For the set of 55 ADRB2 we use for comparison, we have added a supplementary table (Supplementary file 1) with a list of their origin species.

Additionally, we have scanned the text for any instances where referencing conservation was vague and added clarification throughout the manuscript (see below):

Subsection: “Measurement of mutant activities and comparison to evolutionary metrics”

“Mutational tolerance, the mean activity of all amino acid substitutions per residue at each agonist concentration, is highly correlated to conservation, both across species for the β_2_AR (Figure 3—figure supplement 1A; Spearman's ρ = -0.743; 55 orthologs identified from the OMA Database, see Materials and methods), and across all Class A GPCRs (Spearman's ρ = -0.676; Figure 3A, Figure 3—figure supplement 1B)(Altenhoff et al., 2018; Capra and Singh, 2007; Hopf et al., 2017) at EC_100_.”

Subsection: “Mutational tolerance stratifies the functional relevance of structural features”

“More broadly, these ECL1/TM3 positions conserved across Class A GPCRs could serve as candidate sites for introducing thermostabilizing mutations.”

Figure 2 Legend

“Conservation track (Cons.) displays the sequence conservation of each residue across 55 β_2_AR orthologs from the OMA database(Capra and Singh, 2007).”

Figure 6 Legend

“Sequence conservation of extracellular loop 1 (ECL1) and the extracellular interface of TM3 (202 Class A GPCRs with a disulfide bridge between TM3 and ECL1).”

Figure 5—figure supplement 2 Legend

“As predicted, the highly conserved, across species and class A GPCRs, W158^4x50^ is the most constrained residue.”

One of the major findings is the identification of the conserved EL2 motif WxxGxxxC, proposed to work as a 'latch'. It would have been very interesting indeed to see this hypothesis tested in some way, but hopefully this will come in the near future.

We certainly agree with the reviewers that this is an interesting, understudied aspect of GPCR biology. In particular, the identification of the latch highlights the ability of our approach to point towards interesting biology in an unbiased manner. We look forward to and hope to be involved in future investigations of the latch.

Likewise, the observations that distal mutations in the N-terminus and C-terminus lead to constitutive activity invites further studies. Is it possible to say something about this based upon mutagenesis already reported in the literature for beta-2 or other class A receptors? It appears likely that especially the N-terminal mutations may compromise biosynthesis and handling in ER and Golgi, why the caveat should be mentioned early in the manuscript that differences in expression level may explain the observed output results.

We originally hinted at the connection between surface expression and constitutive activity in relation to mutations at the termini. However, we have clarified our statement to reflect the limitations of the assay and interpretability of data (see below):

“Concentration at the termini is unsurprising, as these regions have known involvement in surface expression and our current assay does not discriminate between increased signaling potency and expression (see Discussion; Dong et al., 2007).”

In addition, we have added an anecdote describing a mutation in the N termini of another Class A GPCR, the Melanocortin 4 Receptor (MC4R), that has a constitutively active mutation in the N terminus that increases cAMP signaling while maintaining wildtype surface expression (see below):

“However, there are cases of constitutively active mutations in the N terminus that increase signaling potency without affecting surface expression, such as T11S of the melanocortin 4 (MC4R) (Lotta et al., 2019).”

The comparison of the latch with two receptors that are closely related to each other (opioid kappa and mu) seems a bit superfluous. It would be more interesting if a few completely different peptide receptors were included in the comparison.

We thank the reviewer for the feedback. The structural renderings in panel A show different receptors. In the comparison presented in panel B, we have shown an overlay of inactive state and active state structures, to highlight that the structural latch is present in both inactive as well as active states. As suggested by the reviewer, we have now exchanged the kappa opioid receptor with a different peptide receptor (Angiotensin receptor AT1R) for comparison.

The observation of the EL1 latch has some precedence, see review by Hulme in TIPS, 2013, Figure 3A. Please check if this should be cited.

We have added this citation to the section where we introduce the conserved, previously observed contacts between the Trp and disulfide bond. See below:

“Furthermore, W99^23x50^ is proximal to the disulfide bond C106^3x25^-C191^45x50^, an important motif for stabilization of the receptor’s active state(Noda et al., 1994; Dohlman et al., 1990; Hulme, 2013).”

ADRB2 variants were synthesized in oligonucleotide microarrays split into 8 segments and integrated into the cell line. Additional details on the scheme and numbers/statistics on coverage, library wt representation, and evenness would be important to discuss and show – especially for reproducibility. (Rubin et al., Genome Biology, 2017).

– We have added the total number of reads per condition to the Materials and methods.

– As we did not explicitly design WT mutations into our oligo library, the WT representation is extremely low (only 22 barcodes mapping to synonymous mutations passed our initial filters). This low rate of synonymous mutations is to be expected as they could only come from synthesis or PCR errors that reverted mutant codons back to the WT sequence. Given the lack of barcodes and the spurious nature by which they were generated, we elected to normalize to the frameshifts.

– Figure 1—figure supplement 1H shows the distribution of barcodes per variant for the in one repeat of the EC100 condition which is representative of the other conditions.

– Figure 1—figure supplement 1G is a heatmap showing the barcodes per variant for each mutation in our library. The distinct differences in coverage between sections of the protein correspond to the various fragments we mutagenized.

The authors conduct the DMS experiment under four different isoproterenol conditions and normalize measurements to forskolin treatments. Experimental details on the forskolin activation in their assay or reference for this treatment would aid in interpreting the normalization approach.

The use of Forskolin as a normalization technique is novel and useful for our specific application. We have described it in the main text as follows:

“We normalized these measurements against forskolin treatment, which induces cAMP signaling independent of the β2AR. […] Finally, we define activity as the ratio of this value to the mean frameshift (Materials and methods).

Experimental details for the application of the Forskolin normalization are described in the subsection “Multiplexed Variant Functional Assay Agonist Stimulation, RNA Preparation and Sequencing” in the Materials and methods section.

We have added a reference for a review, Forskolin as a Tool for Examining Adenylyl Cyclase Expression, Regulation, and G Protein Signaling (Insel and Ostrom 2003) to provide context on the use of forskolin for studying cAMP signaling and GPCRs.

What exactly distinguishes the globally intolerant clusters (clusters 1 and 2) in Figure 4? It seems there is a tighter range of activity to isoproterenol in cluster 2 than in 1 for all mutations and chemical properties, but does this get ranked differently than cluster 1?

The differences between clusters 1 and 2 seems to be driven by what the WT residue was originally. For example, in cluster 1, ~55% of residues were originally aromatic (F, W, or Y) and ~41% were originally hydrophobic (I, L, V, or M) opposed to ~7% and ~12% in cluster 2. Alternatively, cluster 2 residues were primarily (~35%) nucleophilic (S or C).